# Role of rosuvastatin and pitavastatin in alleviating diabetic cardiomyopathy in rats: Targeting of RISK, NF-κB/ NLRP3 inflammasome and TLR4/ NF-κB signaling cascades

**Dalia O. Saleh[1], Nesma M.E. Abo El Nasr[1]\*, Marawan A. Elbaset[1], Marwa E. Shabana[2], Tuba Esatbeyoglu [3]\*, Sherif M. Afifi[4], Ingy M. Hashad[5]**

**1** Pharmacology Department, Medical Research and Clinical Studies Institute, National Research Centre, Cairo, Egypt, **2** Pathology Department, Medical Research and Clinical Studies Institute, National Research Centre, Giza, Cairo, Egypt, **3** Department of Molecular Food Chemistry and Food Development, Institute of Food and One Health, Gottfried Wilhelm Leibniz University Hannover, Hannover, Germany, **4** Department for Life Quality Studies, Rimini Campus, University of Bologna, Rimini, Italy, **5** Biochemistry Department, Faculty of Phamacy and Biotechnology, German Universty in Cairo, Cairo, Egypt

\* nassoma84@yahoo.com (NM.E.AEN); esatbeyoglu@foh.uni-hannover.de (TE)

## Abstract

Diabetic cardiomyopathy (DCM) is a serious outcome of type II diabetes mellitus (T2DM) and a key contributor to high morbidity and death in diabetic individuals. The current research is intended to elucidate and compare the therapeutic benefits of rosuvastatin (RVS) and pitavastatin (PTS) in mitigating DMC-induced in rats and exploring the possible underlying molecular signaling pathways. DCM was prompted by feeding rats a high-fat/fructose (F/Fr) diet for eight weeks with a sub-diabetogenic dose of streptozotocin (35 mg/kg; i.p) injection at week seven. All rats were allocated into four groups: a normal control group, a DCM-induced positive control group, the RVS group of DCM-induced rats that were treated once daily with 10 mg/kg of RVS, and the PTS group of DCM rats that were treated with 0.8 mg/kg of PTS. Rats were given the treatments orally for four consecutive weeks. The outcome of the existing work discovered that RVS and PTS significantly improved T2DM-associated DCM, as evidenced by the amelioration of glucose, lipids, cardiac markers, ECG parameters, and redox status. Considering the relationship between oxidative stress and inflammation, this attenuation was evidenced by the downregulation of redox, inflammatory, and cellular fibrotic cascades, namely RISK, NF-κB/NLRP3 inflammasome, and TLR4/NF-κB signaling pathways. Additionally, the histopathological examinations confirmed these structural alterations in the myocardium. Besides, RVS and PTS diminished the expression of caspase-1 assessed by immunochemical staining. In summary, the present study demonstrated that RVS and PTS mitigated the metabolic abnormalities associated with T2DM-induced DCM.

**Data availability statement:** All relevant data are within the manuscript.

**Funding:** The author(s) received no specific funding for this work.

**Competing interests:** The authors have declared that no competing interests exist.

## 1. Introduction

Diabetic cardiomyopathy (DCM) is a significant pathological feature of type II diabetes mellitus (T2DM) characterized by cardiomyocyte hypertrophy, inflammation, perivascular fibrosis, and apoptosis [1,2]. Additionally, it is characterized by decreased glucose utilization and abnormal lipid metabolism in cardiomyocytes, leading to pathological alterations that subsequently result in myocardial remodeling and insufficiency [3,4].

Inflammatory reaction and generation of reactive oxygen species are significant aspects that contribute significantly to the development of DMC and are involved in initiating and progressing pathological hypertrophy remodeling which eventually lead to heart failure [5].

Importantly, long-lasting hyperglycemia fosters the creation of advanced glycation end-products (AGEs). The engagement of AGEs with their receptor enhances the generation of reactive oxygen species (ROS) and triggers the activation of the nuclear factor kappa B (NF-κB) pathway, which in turn triggers inflammatory processes. These processes are mediated by the pro-inflammatory toll-like receptor-4 (TLR-4) [5], thus lead to the transcriptional upregulation of inflammatory cytokines. Furthermore, the nucleotide-binding domain, leucine-rich-containing family, pyrin domain-containing-3 (NLRP3) inflammasome, interleukin 1-beta (IL-1β) as well as interleukin-18 (IL-18) collectively stimulate the transformation of fibroblasts into myofibroblasts and promote collagen synthesis. As a result, this cascade ultimately leads to cardiac fibrosis, remodeling, and dysfunction [6,7].

The NLRP3 inflammasome is a crucial multiprotein complex that exerts a crucial function in regulating the innate immune system and inflammatory signaling pathways. Elevated blood sugar levels, known as hyperglycemia, trigger the activation of NLRP3, which in turn promotes the activation of procaspase-1 which leads to the cleavage of caspase-1. Consequently, the cleaved caspase-1 expedites the maturation of IL-1β as well as IL-18, thereby provoking chronic inflammation-mediated programmed cell death, including pyroptosis, which contributes to the development of DCM [8]. As a result, targeting the regulation of NLRP3 inflammasome pathways represents a promising strategy to combat DCM.

On the other hand, reperfusion injury salvage kinase (RISK) pathway exerts a notable influence role in the context of DCM, which primarily involves a series of signaling events that occur during the reperfusion phase of ischemic injury to the heart. However, its relevance extends to DCM due to the shared mechanisms involved in both conditions [9]. The RISK pathway consists of several protein kinases, including protein kinase B (AKT), extracellular signal-regulated kinase (ERK) and glycogen synthase kinase-3β (GSK-3β). These kinases participate in several biological functions, including cell survival, control of apoptosis, and metabolism. Researchers continue to research ways to use this mechanism for treatment of diabetic heart diseases [10]. Notably, Akt is a kinase that suppresses inflammatory cytokines and free fatty acids [11]by inhibiting caspase-3, cardiomyocytes are protected against apoptosis. Furthermore, GSK-3β is the major substrate of Akt, and it dephosphorylated the GSK-3 to be active. Its activation is linked to apoptosis and the development of DCM.

GSK-3β remains active, resulting in the activation of caspase-3 and consequent cardiac death in patients with chronic diabetes [12].

Statins, on the other hand, are inhibitors of hydroxymethylglutaryl coenzyme A (HMG-CoA) reductase with a lipid-lowering impact and cardiovascular benefits. Statins serve an essential role in cardio-protection by efficiently decreasing low-density lipoprotein (LDL) cholesterol levels, hence lowering the risk of atherosclerosis and cardiovascular disease. In addition, statins possess anti-inflammatory characteristics, enhance endothelial function, and stabilize atherosclerotic plaques, hence reducing their propensity to rupture and produce unexpected blockages. Statins are routinely used to prevent repeated heart attacks and strokes, as well as other cardiovascular risk factors [13].

Rosuvastatin (RVS) and pitavastatin (PTS) are statin members that utilize several pharmacological activities and pleiotropic effects, including antioxidant, anti-inflammatory, and cardioprotective activities [14,15]. Both RVS and PTS have shown potent effects in modulating various signaling pathways, including the RISK pathway, NF-κB/NLRP3 inflammasome, and TLR4/NF-κB signaling pathways. These pathways are critical in addressing the inflammation, oxidative stress, and fibrosis that contribute to disease progression in DCM [16]. Moreover, RVS and PTS have been shown to exhibit strong lipid-lowering effects and may have unique anti-inflammatory properties compared to atorvastatin, making them particularly relevant for our research [16,17]. Therefore, the goal of the current study is to investigate the possible therapeutic actions of RVS and PTS in attenuating the cardiomyopathy complication associated with T2DM induced in rats using the F/Fr/STZ model. Furthermore, this research aims to offer valuable insights into the mechanisms involved in the cardioprotective effects of the statins RVS and PTS through the modulation of the RISK pathway, as well as the NF-κB/NLRP3 inflammasome and TLR4/NF-κB signaling pathways. To the best of our knowledge, this study is the inaugural identification of the beneficial impact of RVS or PTS in milieu DCM via curbing the ROS/TLR/NF-κB as well as regulating AKT/GSK3β signaling cascades.

## 2. Materials and methods

### 2.1. Animals

For this study, a total of 32 adult male albino rats from a local strain were selected as animal model. The animals were housed in the animal house of the National Research Centre, Giza, Egypt. They were kept in separate cages under standard environmental conditions (24 ± 2 °C, relative humidity 55 ± 15%, and 12-hour (h) light-dark cycle). The animals were kept for at least one week for adaptation before being subjected to laboratory experiments. All procedures and experiments have been approved by the "Medical Research Ethics Committee (MREC)" at the "National Research Centre's ethical standards" for the care and use of experimental animals "MREC" (Reg. No. 112311122022).

### 2.2. Drugs and chemicals

Streptozotocin (STZ) was bought from Sigma-Aldrich Co., St Louis, USA. Rosuvastatin (RVS) and pitavastatin (PTS) were bought from Astra Zeneca (Cairo, Egypt) and Mash Premiere (Cairo, Egypt), individually. The highest analytical grade was chosen for all other substances that were employed.

### 2.3. Induction of cardiomyopathy

Rats were fed a high fat/fructose (F/Fr) diet that had unrestricted access to high fat diet (HFD, 1% cholesterol powder, 14% saturated animal fat, 21% protein, 60% carbohydrate, 3% fibers and 1% vitamins and minerals) and 20% (w/v) of fructose solution for eight weeks. At week seven, the overnight fasting rats were injected intraperitoneally with a single sub-diabetogenic dosage of STZ (35 mg/kg; i.p.) in citrate buffer (pH 4.8, 0.09 M) and given a glucose solution (5% w/w) for the first 24 hours [18,19]. The T2DM model was verified by testing fasting blood glucose levels with an Accu-Check glucometer one week after STZ injection (Roche, Mannheim, Germany). Only rats with blood glucose levels ranging from

250 to 350 mg/dl were included in the current study. The animals were afterwards given a regular diet for the remainder of the experiment.

## 2.4. Study design

Rats confirmed to have diabetes were randomly assigned to three groups: an untreated diabetic cardiomyopathy (DCM) group, a DCM group treated orally with (RVS, 10 mg/kg) or (PTS, 0.8 mg/kg) once daily for four weeks. In addition to another group was fed a standard diet with free access to water, which served as a normal control. At week seven, these rats received a citrate buffer injection (1 ml/kg, i.p.). Blood glucose levels were measured at two time points: initially, one week after STZ administration, to confirm diabetes induction for group allocation; and again at the end of the experimental period to assess the maintenance of hyperglycemia and treatment effects.

At the end of the experiment, electrocardiography was performed, and blood samples were collected from overnight fasted rats under light anesthesia. The serum was obtained by centrifuging the samples at 1800 x g for 10 min at 4 °C and used for biochemical analyses. The rats were then sacrificed under $CO_2$ euthanasia, followed by cervical dislocation, and the heart and aorta tissues were dissected.

Some of the tissue from each group was fixed in 10% formalin buffer for 24 h for histological examination. The remaining tissues were homogenized in ice-cold phosphate buffer and centrifuged at 1800 x g for 15 min at 4 °C. The supernatant was used for additional biochemical analyses.

## 2.5. Electrocardiography (ECG)

Electrocardiography was recorded one day prior to scarification, rats were anesthetized with ketamine/xylazine [20–22], and ECGs were documented through the aid of ECG "Powerlab module which consists of Power-lab/8sp and Animal Bio-Amplifier, Australia, in addition to Lab Chart 7 software with ECG analyzer" [23–25].

## 2.6. Biochemical analyses

**2.6.1. Serum biochemical parameters.** Blood glucose levels and lipids were estimated using Total cholesterol (TC) colorimetric kit (Cat. #NS230001) and Triglycerides (TGs) colorimetric kit (Cat. # NS314001) (Salucea, Netherlands).

**2.6.2. Cardiac biochemical parameters.**

**2.6.3.1. Cardiac colorimetric analysis** Cardiac homogenates were used in the colorimetric estimation of reduced glutathione (GSH) (Biodiagnostic, GR 25 11, Egypt) and malondialdehyde (MDA) (Biodiagnostic, MD 25 29, Giza, Egypt) contents. In addition, phosphorylated-serine/threonine protein kinase (P-AKT) (MyBioSource, MBS775153, San Diego, USA), phosphorylated-Glycogen synthase kinase-3 (P-GSK-3 β, MyBioSource, MBS7306987, San Diego, USA), cardiac Troponin (SunLong Biotech Co, SL0121Mo, Yuhang, China), Interleukin-1 beta (IL-1β) (Elabscience, E-EL-R012, Texas, USA), and NLRP3 (OKCD04232−48 Lot# KD0290, San Diego, USA) levels were all measured in heart homogenates using the ELISA technique according to manual instructions.

**2.6.3.2. Cardiac gene expression of NF-κB and TLR4** Total RNA was extracted from the heart tissues of male rats using TRIzol Reagent (Invitrogen, California, USA) following the standard protocol described previously [26]. The extracted RNA aliquots were immediately used for reverse transcription (RT). The RT process involved an initial incubation at 25 °C for 10 min, followed by incubation at 42 °C for 1 h. The reaction was concluded by heating at 99 °C for 5 min. Subsequently, the tubes containing the RT preparations were rapidly cooled on ice before being employed for cDNA amplification through real-time polymerase chain reaction (RT-PCR).

- *Quantitative Real Time-Polymerase Chain Reaction (qRT-PCR)*

StepOne™ Real-Time PCR System from "Applied Biosystems (Thermo Fisher Scientific, Waltham, USA)" was used to estimate the cardiac gene expression of the inflammatory NF-κB and pro-inflammatory TLR-4 markers. "PCR reactions

were set up in 25 µL reaction mixtures containing 12.5 µL 1× SYBR® Premix Ex TaqTM (TaKaRa, Biotech. Otsu, Japan), 0.5 µL 0.2 µM sense primer, 0.5 µL 0.2 µM antisense primer, 6.5 µL distilled water, and 5 µL of cDNA template" [27,28]. The specific primer sequences for the utilized genes are detailed in Table 1. The 2−ΔΔCT method was used to determine the target's relative quantification to the reference gene [29].

## 2.7. Histological examination of the cardiac and the aortic tissues

Aorta and heart samples were rapidly collected from rats in each group. The tissues were rinsed in Aorta and heart samples were promptly collected from rats in different groups, rinsed with normal saline, cleaned of fat, and fixed in buffered 10% formalin. The samples underwent dehydration using ascending serial dilutions of ethyl alcohol, were cleared in xylene, and embedded in paraffin at 56°C in a hot oven for 24 hours. Paraffin blocks were sectioned at a thickness of 4 µm. The sections were mounted on glass slides, deparaffinized, and stained with hematoxylin and eosin for histological examination. Additionally, de-waxed sections of 4 µm thickness were stained with Masson's trichrome for collagen detection. Images were captured using a digital camera (Microscope Digital Camera DP 70, Tokyo) and processed using Adobe Photoshop, Version 8.0.

Quantitative histomorphometric analysis was performed to evaluate the extent and distribution of fibrosis using an Image Analysis System at the Pathology Laboratory in the Medical Research Centre of Excellence (MRCE) unit at the National Research Centre. The analysis utilized the "Leica Qwin DW3000 image analysis system (LEICA Imaging Systems Ltd, Cambridge, England)", consisting of a Leica DM-LB microscope connected to a JVC color video camera and computer system. The software was programmed to quantitatively assess fibrosis. The fibrosis extent was expressed as the percentage of the total examined myocardial area affected. The area percentages were determined by measuring the fibrotic area per microscopic field in micrometer squares.

The measurement of myocardium cell size and aortic wall thickness was performed using image analysis at magnifications of ×400 and ×200, respectively. The diameter of cardiac myocytes was measured for 10 myocytes selected per field across 5 slides, and mean values were calculated from the data for each set of 10 myocytes. For aortic wall thickness, three different areas of the aorta were examined and quantified. The extent of fibrotic tissue was scored after "Masson Trichrome stain" by the semiquantitative as per Galati et al., "visual evaluation of each type of fibrosis: Score 0 when absent, Score + when present in ≤30% of the examined myocardium, Score ++ when present in >30% but <60% of the examined myocardium and Score +++ when present in ≥60% of the examined myocardium" [30].

## 2.8. Immunohistochemistry analysis of cardiac and aortic caspase-1

One section from the tissue block of each group was mounted on positively charged glass slides for **caspase-1** staining. Antigen retrieval was conducted by heating the tissue sections in citrate buffer using a steamer for a duration of 20 min. The sections were then immunostained overnight at 4 °C with a primary antibody against caspase-1. Following incubation with a secondary antibody, immunohistochemical staining was developed using diaminobenzidine as the chromogen.

Table 1. Primers sequence used for *qRT-PCR*.

| Gene | Primer sequence | NCBI Reference |
|------|-----------------|----------------|
| *NF-κB* | F: TTCCTGCTTACGGTGGGATT | NM_001276711.1 |
| | R: CCCCACATCCTCTTCCTTGT | |
| TLR4 | F: TATCGGTGGTCAGTGTGCTT | NM_019178.2 |
| | R: CTCGTTTCTCACCCAGTCCT | |
| GAPDH | F: AAT GGG AAG CTT GTC ATC AA | NM_017008.4 |
| | R: TAC TTG GCA GGT TTC TCC AG | |

## 2.9. Statistical analysis

Values are presented as mean ± SEM of 6 animals. The statistical analysis was achieved using GraphPad software. V7, San Diego, CA, USA. The differences between groups were tested for significance using one-way ANOVA, followed by a post-hoc test (Tukey's multiple comparisons test) for the parametric test. Furthermore, the Kruskal-Wallis as a post-hoc was performed for non-parametric pathological scores where $p \leq 0.05$. If there were any missing data before analysis, we handled it using listwise deletion, wherein cases with missing values were excluded from the respective analyses. The effect size of each statistical test was reported in the supplementary data (S1 File).

# 3. Results

### 3.1. Effect of RVS and PTS on metabolic biomarkers against T2DM-induced DMC in rats

The results indicated that rats with DCM demonstrated higher serum glucose and lipids (TC and TG) levels, by 2.7-, 2.0-, and 2.1-fold, as compared to the normal control group. On the other hand, the RVS and PTS groups exhibited an improvement in serum glucose by 41% and 33% in relation to the DCM group. Furthermore, treatment with RVS and PTS led to significant declines in elevated TC and TG levels, by 61% and 58%, and by 57% and 45%, respectively, compared to the DCM control group (Fig 1).

### 3.2. Effect of RVS and PTS on ECG alterations against T2DM-induced DMC in rats

As illustrated in Fig 2, the alteration of cardiac contractility of DCM rats was evidenced by a 43% increase in heart rate, a 30% shortening of the R-R interval, and a 40% rise in the R-wave amplitude compared to normal control. In contrast, treatment with PTS or RVS mitigated these changes, manifested by reducing the heart rate by 16% and 17%, respectively. Additionally, both treatments led to an elongation of the R-R interval by 19% and 21%, respectively, and a reduction in the R-wave amplitude by 40% and 35%, compared to DCM rats. These findings demonstrate that PTS and RVS help restore the altered cardiac function.

In terms of cardiac conductivity and rhythmicity, Fig 3 highlights significant disruptions in DCM rats, by a 36% prolongation in the QRS interval and a 90% increase in the QTc interval. Furthermore, there was a notable reduction in both ST height and the PR interval, by 43% and 10%, respectively, compared to the normal control group. These alterations reflect impaired electrical conduction and rhythmicity in DCM. However, PTS and RVS countered these disturbances, evidenced by shortening the QRS interval by 16% and 22%, and the QTc interval by 34% and 38%, respectively. Additionally, both treatments significantly improved the PR interval by 7% and increased ST height by 2.0- and 1.8-fold, respectively, compared to DCM rats. These results indicate that PTS and RVS contribute to the restoration of cardiac conductivity and rhythmicity in the context of DCM.

### 3.3. Effect of RVS and PTS on cardiac oxidative stress-related biomarkers against T2DM-induced DMC in rats

Oxidative stress, characterized by an imbalance between reactive oxygen species and antioxidant defenses, is a critical contributor to the pathogenesis of the disease. As portrayed in Fig 4, DCM induced by the F/Fr/STZ model exhibited a substantial boost in oxidative burden, demonstrated by a notable rise of 1.7-fold in the cardiac content of MDA. Furthermore, the results indicated a decrement in GSH content, by 22% as compared to the normal control group. However, treatment with RVS and PTS revealed a prominent reduction in MDA content in cardiac tissues, by 41% and 39%, respectively, as well as an elevation in GSH content, by 28%.

### 3.4. Effect of RVS and PTS on cardiac NLRP3 inflammasome and pro-fibrotic IL-1 β signaling cascades against T2DM-induced DMC in rats

Reactive oxygen species can act as an upstream trigger for inflammatory cascades. So, we also investigate the inflammatory signaling pathways that are triggered by oxidative stress. Fig 5 presents that the DCM group revealed a marked

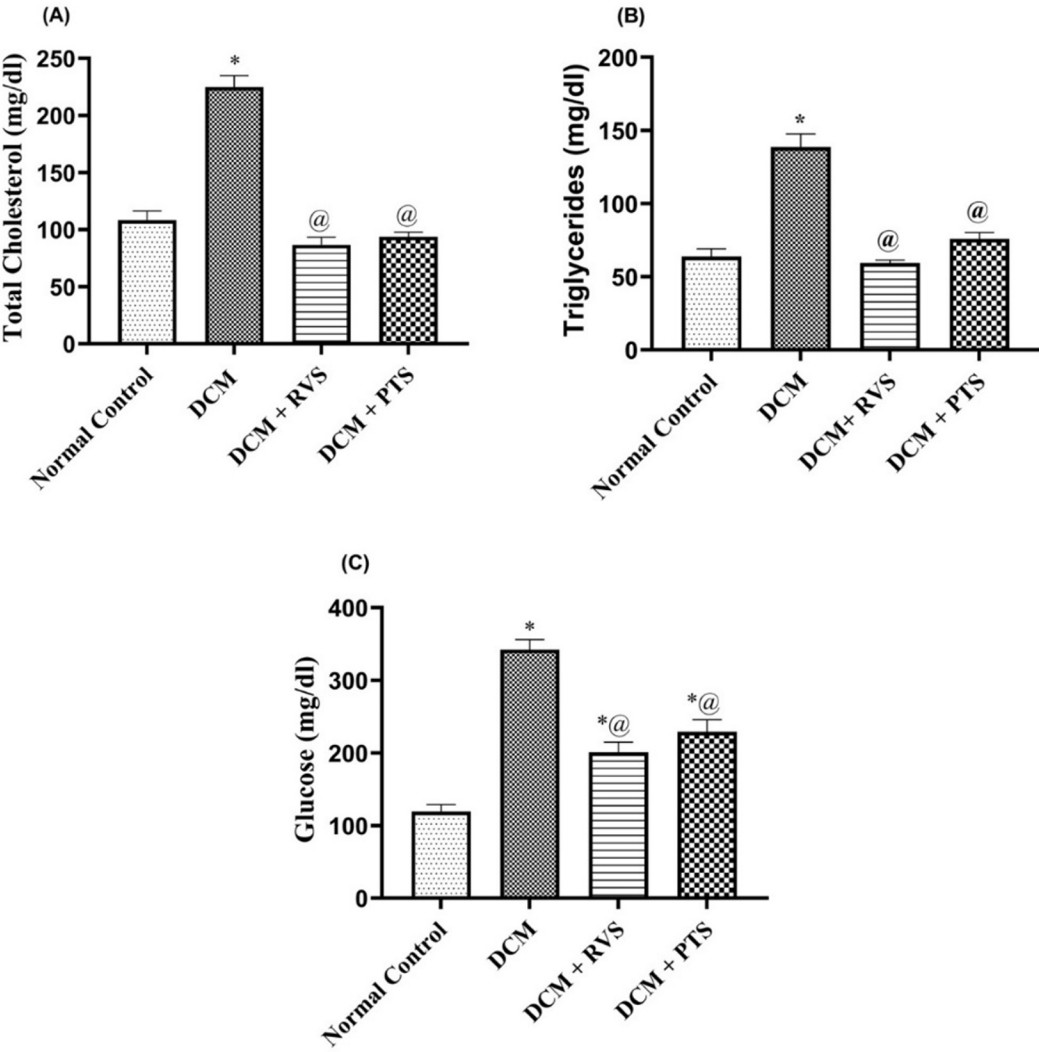

**Fig 1. Effect of Rosuvastatin and Pitavastatin on metabolic biomarkers; Cholesterol (A), Total Triglycerides (B), and Glucose (C) against DCM-induced in Rats.** Each bar represents the mean±SEM of 6 rats. * Vs normal control group & @ vs DCM control group at $p < 0.05$. DCM, diabetic cardiomyo*p*athy; RVS, Rosuvastatin; PTS, Pitavastatin.

elevation in cardiac content of NLRP3 inflammasome as well as a prominent augmentation of cardiac content of IL-1β, by 3.2- and 2.9-fold related to the normal control group, indicating the initiation and exacerbation of diabetic cardiomyopathy due to inflammatory cascades. On the other hand, treatment with RVS or PTS resulted in a 52% or 62% reduction in NLRP3, respectively, as well as a decrease in IL-1β levels, by 28% and 24%, respectively, in respect to the DCM group.

### 3.5. Effect of RVS and PTS on cardiac markers of inflammatory signaling cascades against T2DM-induced DMC in rats

The data revealed that the F/Fr/STZ model induced significant inflammatory cascades, evidenced by a marked elevation in total Akt levels and a reduction in p-GSK-3β, by 40% and 58%, respectively, compared to the normal group which presented in Fig 5. Conversely, the oral gavage of RVS was able to decrease the raised cardiac content of AKT significantly,

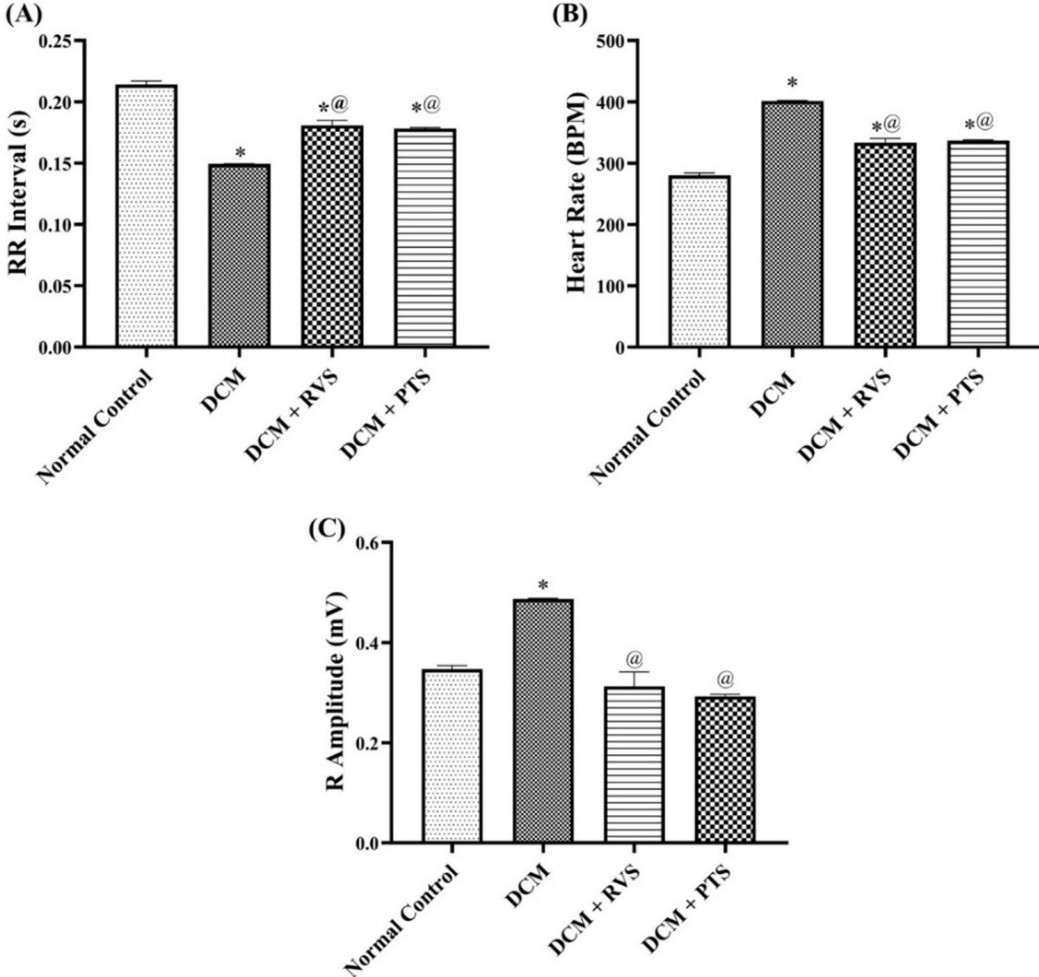

**Fig 2. Effect of Rosuvastatin and Pitavastatin on Cardiac Contractility measurements; RR Interval (A), Heart Rate (B), and R Amplitude (C) against DCM-induced in Rats.** Each bar represents the mean ± SEM of 6 rats. * Vs normal control group & @ vs DCM control group at p < 0.05. DCM, diabetic cardiomyopathy; RVS, Rosuvastatin; PTS, Pitavastatin.

by 15% as compared to the DCM control group. Additionally, RVS and PTS revealed increments in *p*-GSK-3β, by 98% and 77%, respectively, related to the DCM group in Fig 6.

### 3.6. Effect of RVS and PTS on cardiac gene expression of the inflammatory NF-κB and pro-inflammatory TLR-4 markers against T2DM-induced DMC in rats

In Fig 7, the DCM rats evoked an increase of 6-fold of the cardiac inflammatory marker NF-κB gene expression, which induces the inflammatory events mediated by the pro-inflammatory TLR-4 thus resulting in a significant upsurge in cardiac TLR-4 gene expression, by 4.5-fold of the normal control group. RVS and PTS significantly reduced the elevated gene activity of NF-κB, by 32% and 52%, respectively, as well as the diminution of the TLR-4 gene expression by 37% and 48%, respectively, in respect to the DCM group. These findings suggest that treatment with RSV or PTS has a potential therapeutic effect in the mitigation of injurious inflammatory events by halting the inflammatory cascades and subsequently safeguarding against the cardiovascular complications associated with diabetes.

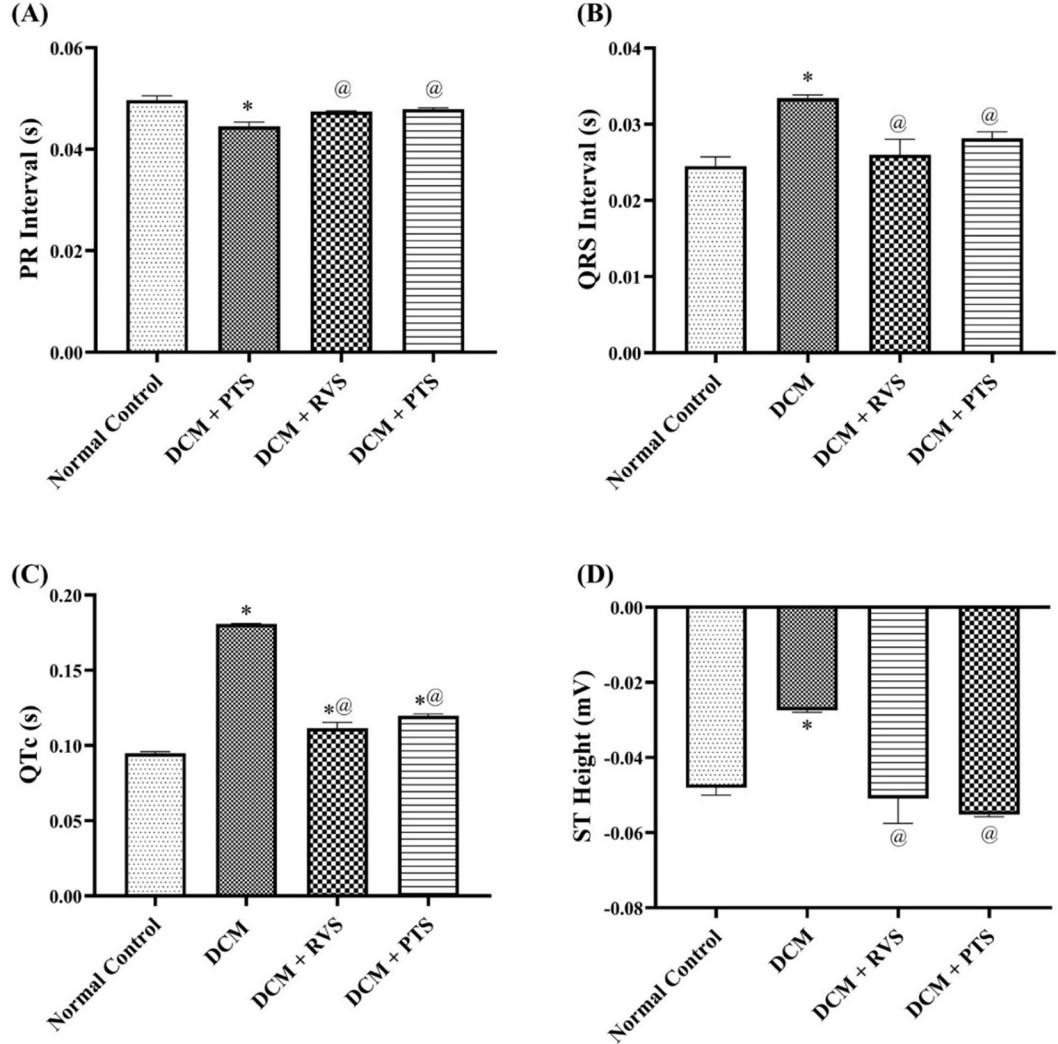

**Fig 3. Effect of Rosuvastatin and Pitavastatin on Cardiac Conductivity and Rhythmicity measurements PR Interval (A), QRS Interval (B), QTc (C), and ST Height (D) against DCM-induced Rats.** Each bar represents the mean ± SEM of 6 rats. * Vs normal control group & @ vs DCM control groupat p < 0.05. DCM, diabetic cardiomyopathy; RVS, Rosuvastatin; PTS, Pitavastatin.

### 3.7. Effect of RVS and PTS on cardiac troponin against T2DM-induced DMC in rats

Likewise, diabetic rats exhibited a notable increase in cardiac troponin levels, rising from 20.24 to 38.66 mg/tissue protein compared to the normal control group. However, RSV and PTS treatment resulted in marked improvements in cardiac troponin, by 37% and 33%, respectively, as matched to the DCM group, as revealed in Fig 8.

### 3.8. Effect of RVS and PTS on histopathological examinations in T2DM-induced DMC in rats

Histopathological examinations revealed normal myocardial architecture in the control group, as illustrated in Fig 9. Hematoxylin and eosin staining shown regular histological heart architecture with myofibrils and muscle bundles without cellular infiltration and normal vasculature (Fig 9A). On the other hand, the F/Fr/STZ-induced DCM model exhibited a marked myocardial accumulation of infiltrating inflammatory cells, along with collagen deposition in cardiac myocytes, indicative of

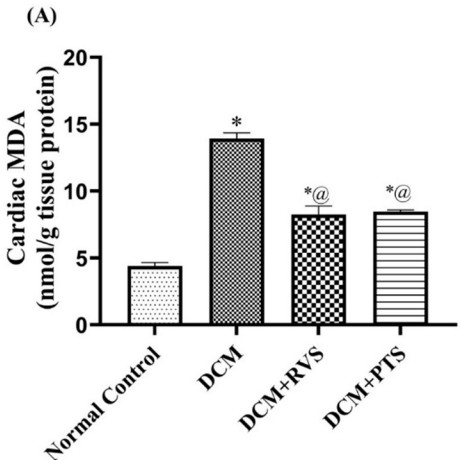
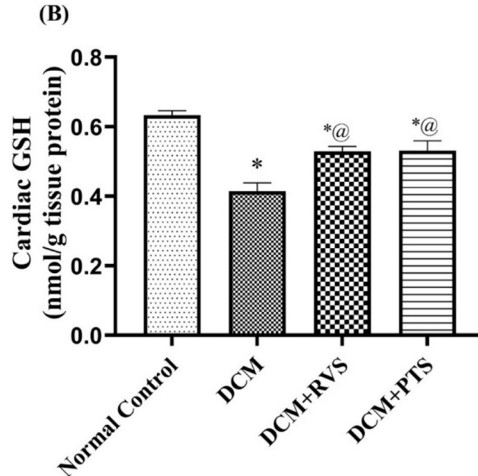

**Fig 4. Effect of Rosuvastatin and Pitavastatin on oxidative stress-related biomarkers; MDA (A) and GSH (B) against diabetic cardiomyopathy-induced in Rats.** Each bar represents the mean±SEM of 6 rats. * Vs normal control group & @ vs DCM control group at $p<0.05$. DCM, diabetic cardiomyo*pathy*; RVS, Rosuvastatin; PTS, Pitavastatin.

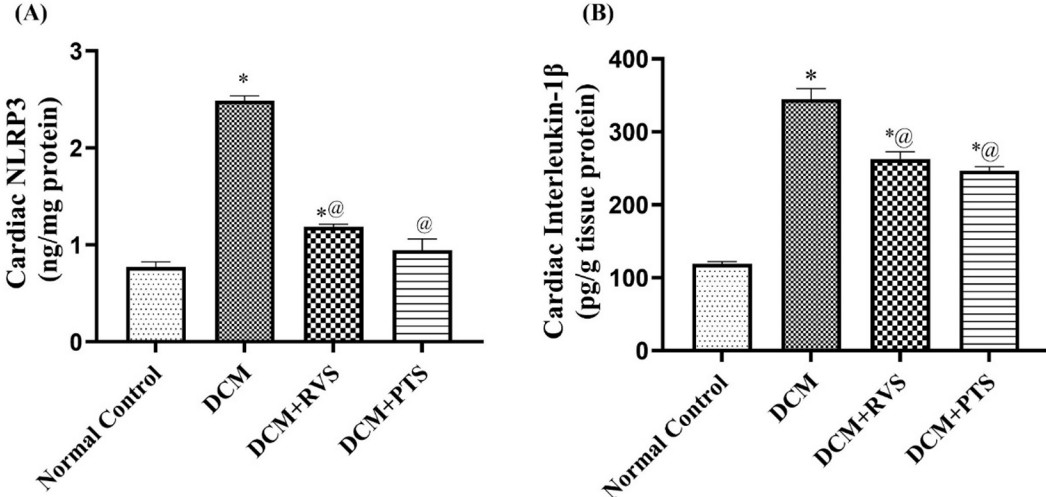

**Fig 5. Effect of Rosuvastatin and Pitavastatin on cardiac NLRP3 inflammasome (A) and pro-fibrotic IL-1 β (B) against diabetic cardiomyopathy-induced in Rats.** Each bar represents the mean±SEM of 6 rats. * Vs normal control group & @ vs DCM control group at $p<0.05$. DCM, diabetic cardiomyo*pathy*; RVS, Rosuvastatin; PTS, Pitavastatin.

evident fibrosis between the cardiac muscle fibers. This was accompanied by fat deposits within the myocardium (Fig 9B). The group of rats that were treated with RSV exhibited few inflammatory cells infiltrate (Fig 9C). In addition, treatment with PTS has shown a normal myocardial architecture, i.e., normal arrangement of myocardial fibers, organized in a parallel and uniform pattern, and the size of cardiac myocytes was normalized without signs of hypertrophy or atrophy (Fig 9D). The treated groups indicated a myocardial fiber arrangement similar to the healthy control group, though with slight irregularities in fiber alignment. The cardiac myocyte size was slightly larger than the normal group but still smaller than in the DCM group, indicating restoration of myocardial structure.

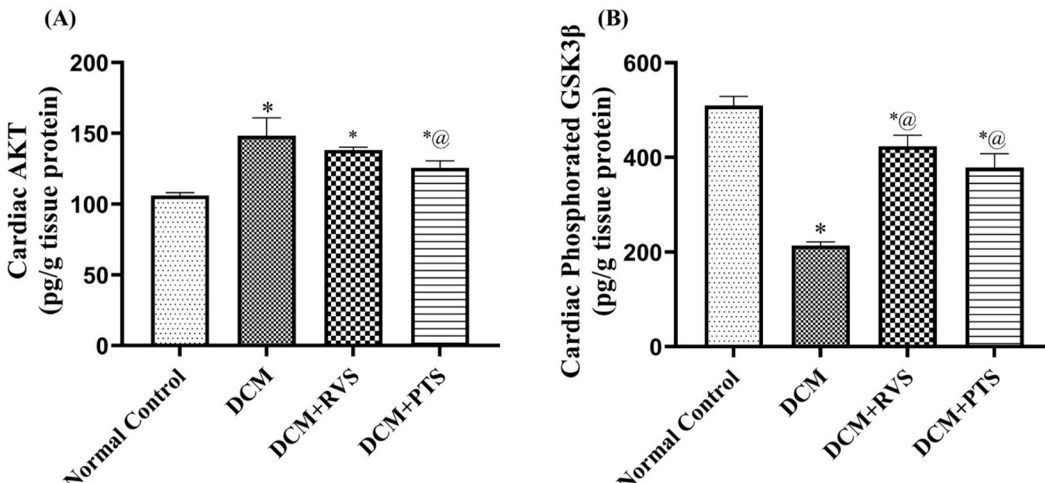

**Fig 6. Effect of Rosuvastatin and Pitavastatin on cardiac Akt (A) and p-GSK-3 β (B) against diabetic cardiomyopathy-induced in Rats.** Each bar represents the mean±SEM of 6 rats. * Vs normal control group & @ vs DCM control group at $p<0.05$. DCM, diabetic cardiomyo*p*athy; RVS, rosuvastatin; PTS, pitavastatin.

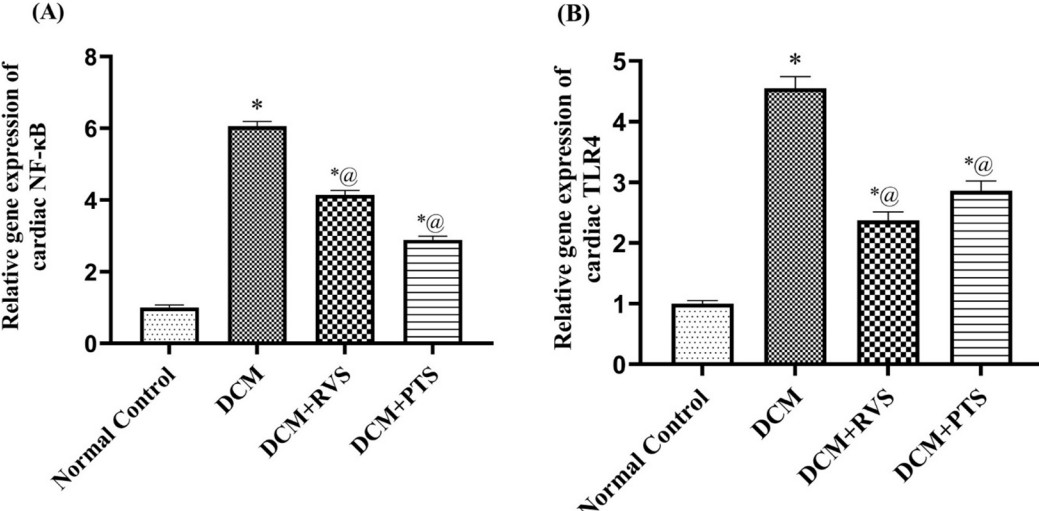

**Fig 7. Effect of Rosuvastatin and Pitavastatin on cardiac gene expression of NF- κB (A) and TLR-4 (B) against diabetic cardiomyopathy-induced in Rats.** Each bar represents the mean±SEM of 6 rats. * Vs normal control group & @ vs DCM control group at $p<0.05$. DCM, diabetic cardio-myo*p*athy; RVS, rosuvastatin; PTS, pitavastatin.

Masson's Trichrome staining was used to assess fibrosis collagen deposit in the myocardium. This staining technique specifically highlights collagen fibers, which are key markers of fibrosis. In this study, Masson's Trichrome staining revealed increased collagen deposition in the DCM group, indicating the presence of myocardial fibrosis. Fibrosis is an important pathological feature of DCM, as the excessive deposition of collagen disrupts the normal architecture and function of the heart. The accumulation of fibrotic tissue can impair myocardial contractility, promote cardiac remodeling, and ultimately lead to heart failure. Therefore, the identification of fibrosis through Masson's Trichrome

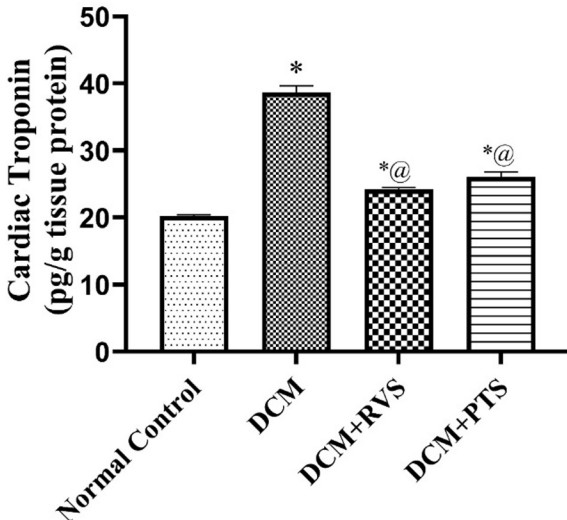

**Fig 8. Effect of Rosuvastatin and Pitavastatin on cardiac troponin against diabetic cardiomyopathy-induced in Rats.** Each bar represents the mean ± SEM of 6 rats. * Vs normal control group & @ vs DCM control group at $p < 0.05$. DCM, diabetic cardiomyo*p*athy; RVS, rosuvastatin; PTS, pitavastatin.

staining underscores the severity of DCM and the potential of treatments, like PTS, to mitigate these pathological changes. Masson's Trichrome staining revealed increased collagen deposition in the DCM group, indicating myocardial fibrosis, a key pathological feature of DCM. Fibrosis disrupts heart function by impairing myocardial contractility and promoting cardiac remodeling, potentially leading to heart failure. The findings highlight the severity of DCM and suggest that treatments like PTS may help mitigate fibrosis, preserve myocardial structure, and improve cardiac function in DCM.

In rats fed a normal diet, cardiac cells were arranged in an orderly fashion (Fig 10A). DCM control group caused by structural disorganization of the myocardial tissue, myocardial diffuse interstitial, and perivascular fibrosis associated with dilated congested blood vessels (Fig 10B). Furthermore, a group of rats treated with RSV exhibited moderate perivascular fibrosis (Fig 10C). Administration of PTS exhibited minimal perivascular fibrosis and normal myocardial architecture, i.e., normal arrangement of myocardial fibers, organized in a parallel and uniform pattern, and the size of cardiac myocytes was normalized without signs of hypertrophy or atrophy (Fig 10D).

Histopathological examination of aortic tissue was conducted, and media thickness was assessed using image analysis of H&E-stained sections at ×200 magnification. The results revealed that aortic tissues from the control non-diabetic group displayed normal histological structures with a mean media thickness of 67.93 μm (Fig 11A, B). However, the aortas of diabetic rats revealed hypertrophied aortic walls with increased thickness (mean = 123.97 μm) due to medial collagen accumulation, likely resulting from increased dietary fat intake and STZ administration (Fig 11C, D). Diabetic rats treated with RVS demonstrated a reduction in media thickness compared to the untreated diabetic group (mean = 98.59 μm) (Fig 11E, F). Notably, treatment with the PTS restored the aortic appearance to near-normal, with a media thickness close to that of the control group (mean = 73.38 μm) (Fig 11G, H).

Additionally, cardiomyocyte diameter was measured at ×400 magnification as illustrated in Fig 11I, showing differences between groups. The mean diameter in the normal control group was 13.42 μm, while it increased to 21.84 μm in the untreated diabetic group. Treatment with RVS reduced the diameter to 16.49 μm, and PTS further reduced it to 15.45 μm.

The extent of fibrosis was quantified in each of the examined hearts, and the average amount of fibrosis is summarized in Table 2. In the Normal Control group, no fibrosis was observed (0). The DCM group exhibited significant fibrosis (+++),

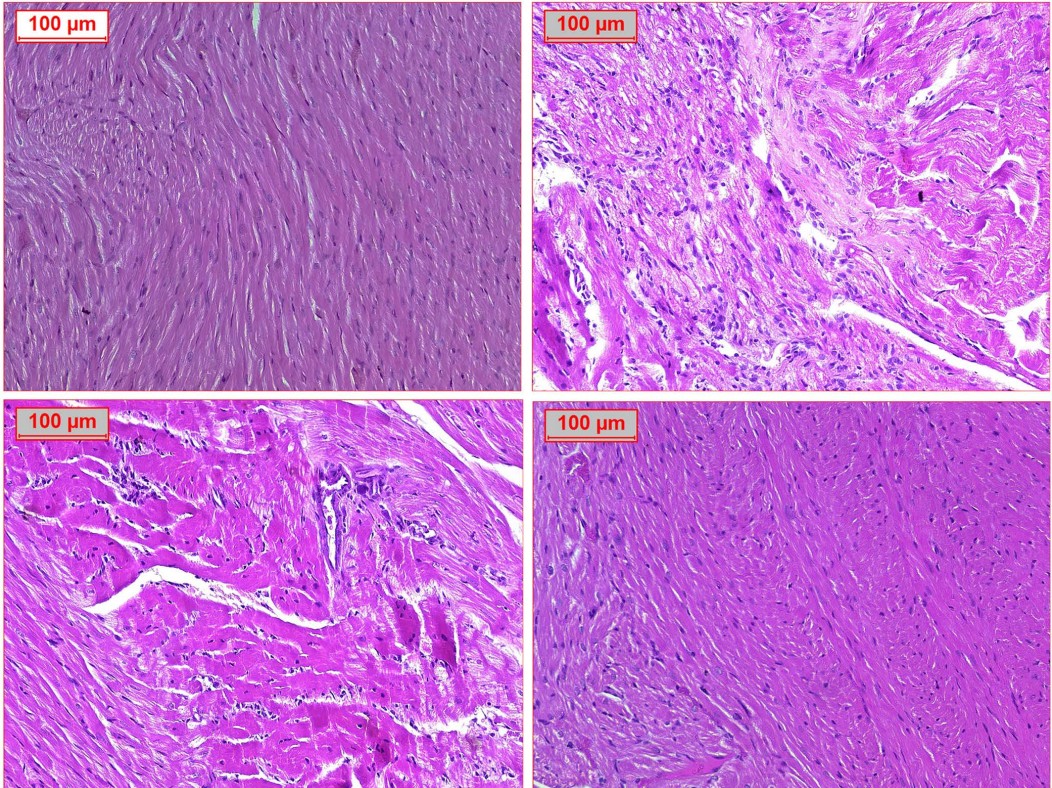

**Fig 9. Effect of Rosuvastatin and Pitavastatin on photomicrograph of H&E stained cardiac tissues of DCM- induced Rats.** Photomicrograph of H&E stained cardiac tissues demonstrated A: normal cardiac muscle bundles. B: diabetic group revealed inflammatory cell infiltrate (thick arrow), fibrosis (star) and scattered fat cells (thin arrow). C: RVS treated group indicated minimal inflammatory cell infiltrate (thick arrow). D: PTS treated group revealed normal cardiac muscle tissue. (H&E x200). The figure illustrates the histological examination of myocardial tissue, including evaluation of inflammatory cell infiltration, fibrosis, and fat accumulation characteristic of DCM. The observed changes were semi-quantitatively assessed and supported by Masson's Trichrome staining for collagen deposition. Treatment with RSV showed partial improvement, while PTS treatment more effectively preserved myocardial architecture, reduced fibrosis, and restored myocyte size closer to the control group values.

while treatment with PTS (DC+PGZ) reduced fibrosis to a mild degree (+), and RVS treatment (DC+RVS) showed moderate fibrosis (++) compared to the DCM group. These findings highlight the varying degrees of fibrosis and underscore the potential effectiveness of the treatments in mitigating fibrotic changes in DCM.

These results were confirmed by caspase-1 immunohistochemical staining, which revealed elevated expression of caspase-1 in the DCM group compared to the normal control, RSV, and PTS-treated groups, as illustrated in Figs 12 and 13. High caspase-1 expression is crucial in the pathogenesis of DCM due to its role in inflammasome activation and the subsequent inflammatory response.

## 4. Discussion

Diabetic cardiomyopathy is a condition characterized by the weakening and enlargement of the heart muscle due to prolonged exposure to high blood sugar levels in individuals with T2DM [31]. This leads to reduced cardiac function and an increased risk of heart failure [30]. The exact underlying mechanisms of DCM remain unclear, but it is believed to be associated with metabolic changes caused by elevated glucose levels, oxidative stress, and the accumulation of advanced glycation end-products (AGEs). DCM may also be linked to other complications of T2DM, such as microvascular disease and autonomic [32].

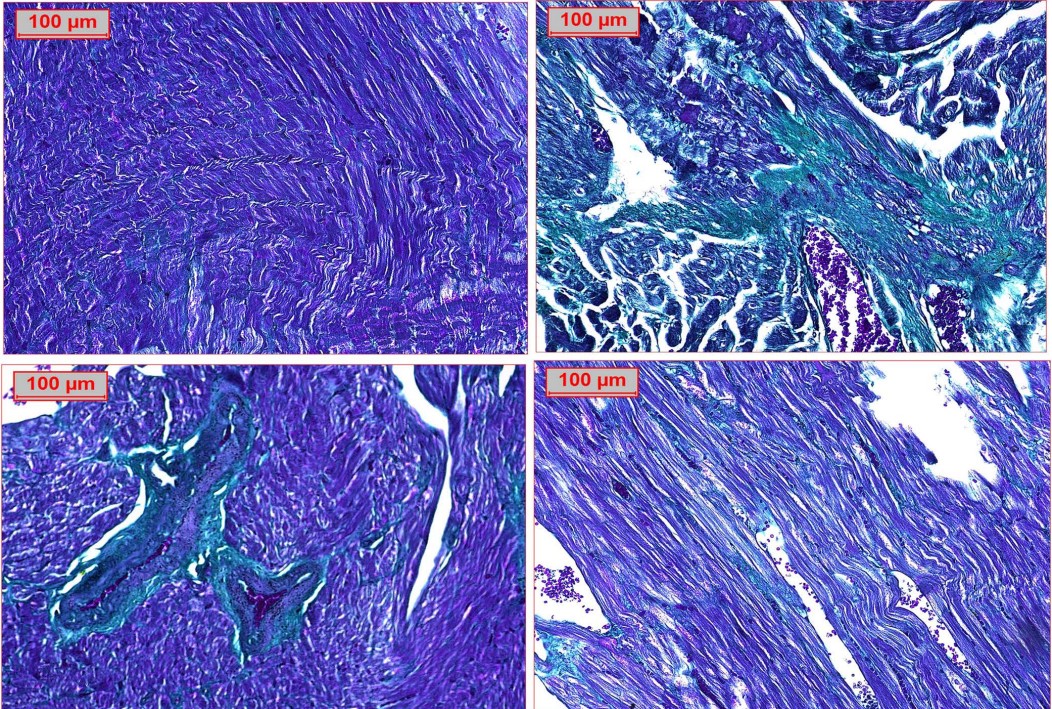

**Fig 10. Effect of Rosuvastatin and Pitavastatin on Photomicrograph of Masson trichrome stained cardiac tissues of DCM- induced Rats.** Photomicrograph of Masson trichrome stained cardiac tissues demonstrated A: normal cardiac muscle bundles. B: diabetic group indicated inflammatory cell infiltrate (thick arrow), fibrosis (star) and scattered fat cells (thin arrow) C: RVS treated group revealed moderate improvement with mild fibrosis. D: PTS treated group revealed marked improvement with minimal fibrosis.

In the present investigation, F/Fr/STZ model involves administering a combination of high-fat diet (F), fructose (Fr), and streptozotocin (STZ) to rats, resulting in the development of T2DM. The results highlight the significant metabolic disturbances associated with DCM, characterized by hyperglycemia and dyslipidemia. Rats with DCM exhibited markedly elevated serum glucose levels (2.7-fold), alongside increased TC and TG levels (2.0- and 2.1-fold, respectively) compared to the normal control group. The study manifested that the diabetic rats displayed significant changes in heart function, *viz.,* depressed ST height, increased R-amplitude, elongated QTc interval, and elevated heart rate. In addition to the increase in blood cardiac marker, troponin, and inflammatory cardiac contents, there is an observed correlation with TLR4/NF-κB and the excessive activation of Akt/GSK3β, all of which are associated with alterations in heart function. These changes were confirmed by histopathological changes and histo-morphometric scores that indicated scarred cardiomyocytes with infiltration of inflammatory cells and scarred tissue. Additionally, aortic changes with prominent fibrosis were observed in diabetic rats.

Our results unequivocally demonstrated significant elongation of QRS and QTc intervals with a reduction of the heart rate in the context of DCM, aligning with other studies that identify these parameters as possible indicators of detrimental cardiac remodeling and electrical instability [23, 24, 33–35]. An extended QRS duration indicates ventricular dyssynchrony and conduction abnormalities, often associated with a worse prognosis due to an elevated risk of arrhythmias and abrupt cardiac death. QTc prolongation indicates a delay in cardiac repolarization and increases the risk of potentially fatal arrhythmias, such as torsades de pointes [34,36]. Our investigation recorded reduced QRS and QTc intervals accompanied by a reduction of the heart rate post-treatment, indicating a potential therapeutic impact that may alleviate the arrhythmic load and enhance ventricular synchronization. These data thus corroborate our hypothesis that the

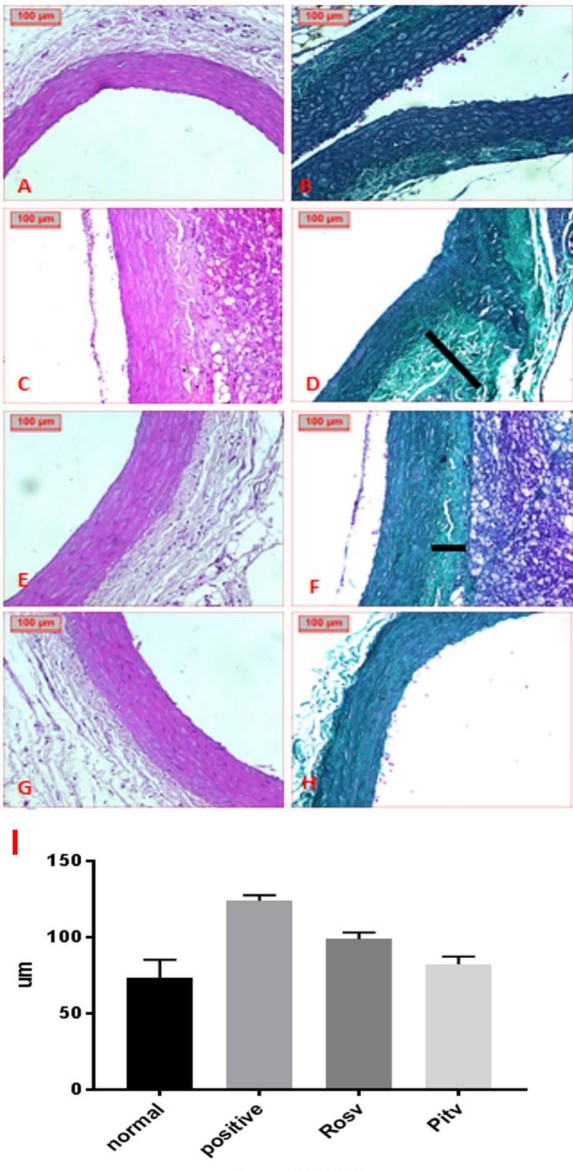

**Fig 11. Effect of Rosuvastatin and Pitavastatin on photomicrograph of H&E and Masson trichrome stained aortic tissues of DCM- induced Rats.** Photomicrograph revealed aortic changes A, B: normal aortic tissue. C, D: diabetic group showed fibrosis (line). E, F: RVS treated group. G, H: PTS treated group (H&E, Masson trichrome, X200).

**Table 2. Effect of Rosuvastatin and Pitavastatin on the heart overall fibrosis in DCM- induced Rats.**

| Groups | Average amount of fibrosis |
|---|---|
| Normal Control | 0 |
| DC rats | +++ |
| DC+PGZ | + |
| DC+RVS | ++ |

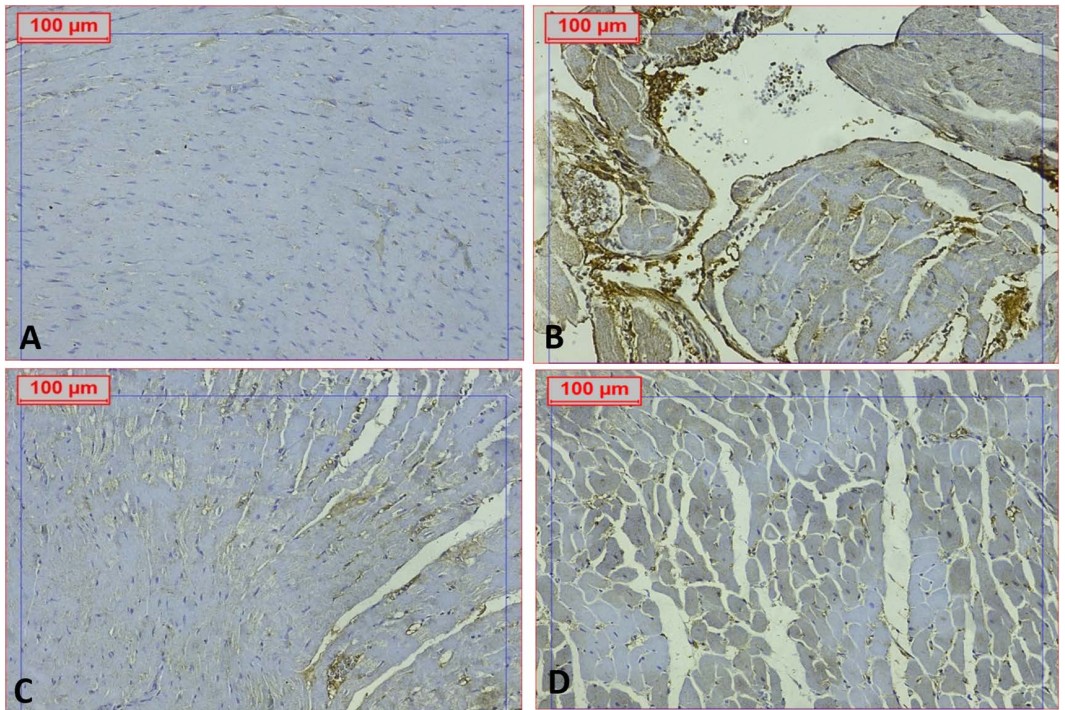

**Fig 12. Effect of Rosuvastatin and Pitavastatin photomicrograph indicated cardiac tissue of DCM-induced rats stained with caspase 1.** Photomicrograph showed cardiac tissue stained with caspase 1: exhibited positive expression in the DCM group, with notable reductions observed in the treated groups. (IHC, caspase 1, X200).

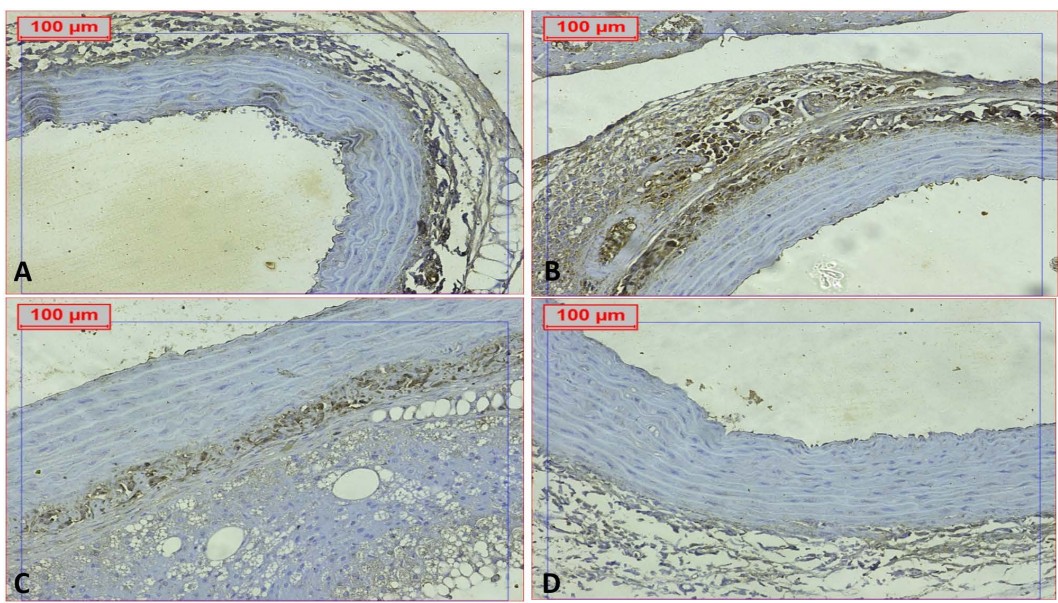

**Fig 13. Effect of Rosuvastatin and Pitavastatin photomicrograph indicated aortic tissue of DCM-induced rats stained with caspase 1.** Photomicrograph displayed aortic tissue stained with caspase 1: exhibited positive expression in the DCM group, with observed reductions in the treated groups (IHC, caspase 1, X200).

normalization of these parameters is associated with improved electrophysiological and structural heart function. This has significant implications for clinical practice, since enhancements in these parameters may lead to decreased morbidity and death in patients with DCM.

Likewise, previous studies on DMC align with the current electrocardiographic and histopathological changes observed in our results [25,37,38]. Left ventricular dilatation and dysfunction during early-diastolic- filling with increased length of isovolumetric relaxation are the structural hallmarks of DMC [39,40] Systolic dysfunction eventually leads to symptomatic heart failure. Stiffened cardiomyocytes, fibrotic cardiac tissue, and enlarged hearts are thought to contribute to cardiac dysfunction [41]. $Ca^{2+}$ release and ventricular tachycardia are exacerbated by this dysfunction, promoting sympathetic tone and increasing sarcoplasmic reticulum activity via hyperstimulation of β-receptors and ryanodine receptors [42]. Inflammatory reactions and cardiac fibrosis further contribute to the disturbance [43].

Moreover, the F/Fr/STZ-induced DCM model exhibited marked myocardial accumulation of infiltrating inflammatory cells, along with collagen deposition in cardiac myocytes, indicative of fibrosis between the cardiac muscle fibers. Fat deposits within the myocardium, commonly associated with myocardial lipid infiltration, a characteristic feature of metabolic disorders like diabetes, were also observed. The accumulation of fat within the myocardium disrupts normal cardiac function, contributing to inflammation, fibrosis, and impaired contractility—key characteristics of DCM [44].

Our study also highlights the underexplored interaction between AGE-induced ROS production and the subsequent activation of the TLR4/NF-κB and NLRP3 inflammasome pathways, which contribute to the acceleration of inflammation and fibrosis in DCM. Overactivation of NLRP3 was accompanied by elevated levels of downstream cytokines, exacerbating myocardial injury by amplifying local inflammatory responses. Furthermore, NLRP3 activation was shown to synergize with the TLR4/NF-κB pathway, as evidenced by the observed upregulation of TLR4 expression and NF-κB nuclear translocation. This cross-talk between NLRP3 and NF-κB perpetuates a pro-inflammatory environment, promoting cardiac fibrosis and structural remodeling [45]. This pathway has been relatively underexplored in previous DCM models. Chronic hyperglycemia increases the production of AGEs, leading to oxidative overload, increased ROS generation, and the amplification of oxidative stress. AGEs interact with the receptor for advanced glycation end products (RAGE) on the cell membrane, triggering downstream signaling cascades such as MAPK, p38, SAPK/JNK, ERK1/2, and JAK/STAT. These pathways activate transcription factors, particularly NF-κB, which induce the expression of pro-inflammatory cytokines and apoptosis-related genes. This cascade promotes inflammation, cell death, and tissue damage, hallmarks of DCM pathology [19]. Interestingly, the DCM group exhibited a significant increase in cardiac NLRP3 inflammasome levels, by 3.2-fold, alongside a 2.9-fold elevation in IL-1β content compared to the normal control group. These findings underscore the activation of inflammatory cascades that exacerbate cardiac damage and contribute to disease progression in DCM.

In the current study, the role of TLR4/NF-κB cascades, triggered by hyperglycemia and ROS, in significantly influencing AGE activation in DCM has been emphasized. AGEs promote oxidative reactions, leading to an increase in ROS production, as evidenced by lipid peroxidation, and a reduction in the cell's defense mechanisms against ROS within cardiac cells [46,47]. These ROS, in turn, contribute to activating various signaling pathways involved in different aspects of DMC, including myocyte hypertrophy [48–50]. Furthermore, ROS can also activate TLR4 signaling pathways associated in interstitial fibrosis, inflammation, and contractile dysfunction, as mentioned in the previous response. Therefore, the interplay between AGE-dependent activation, ROS production, and subsequent signaling pathways have an important role in advancement and progression of DMC [37,51]. NF-κB, as a transcription factor, enhances the expression of pro-inflammatory genes like TLR-4. Hyperglycemia-induced stress activates NF-κB, leading to its translocation to the nucleus, where it upregulates inflammatory genes, creating a feedback loop that sustains chronic inflammation. This process contributes to the pathogenesis of DCM by mediating cardiac inflammation, fibrosis, and structural remodeling, ultimately impairing cardiac function. Understanding this causal pathway highlights the importance of targeting the TLR-4/NF-κB axis for potential therapeutic intervention in DCM [52]. A 6-fold increase in NF-κB, coupled with a 4.5-fold increase in TLR-4, as observed in this study, indicates a significant upregulation of the inflammatory response in DCM. TLR-4 activation

triggers a cascade of downstream proteins, including NF-κB, which enhances the expression of pro-inflammatory genes, sustaining chronic inflammation in the myocardium. This pathway contributes to the pathogenesis of DCM by promoting cardiac inflammation, fibrosis, and structural remodeling, ultimately impairing cardiac function. Additionally, activation of the NLRP3 inflammasome amplifies the inflammatory response through cytokines like IL-1β and IL-18, worsening myocardial injury. Targeting the TLR-4/NF-κB axis and NLRP3 inflammasome presents a potential therapeutic strategy to mitigate inflammation and prevent DCM progression in diabetic patients [53,54]. All these derangements lead to dramatic changes in cardiac tissue, resulting in impairment of cardiac function. The NLRP3 inflammasome plays a vital role in proteolytic reactions and the development of IL-1β to its active form [55], and recent research has revealed that overactivation of NLRP3 is central to the development of metabolic conditions such as T2DM [56].

Interestingly, treatment with RVS and PTS resulted in significant metabolic improvements. Serum glucose levels decreased by 41% and 33%, respectively, in the RVS and PTS groups compared to the untreated DCM group. Both RVS and PTS alleviated cardiac changes, as demonstrated by improved serum cardiac markers, ECG parameters, oxidative status, and reduced inflammatory and fibrotic markers. RVS has been shown to have a beneficial impact on cardiovascular health. Both treatments mitigated the hyperactivity and impaired relaxation observed in DCM, as evidenced by reductions in heart rate (16% and 17%, respectively) and R-wave amplitude (40% and 35%, respectively), along with elongations in the R-R interval (19% and 21%, respectively). Additionally, RVS and PTS improved cardiac electrical conduction and rhythmicity by shortening the QRS interval (16% and 22%) and the QTc interval (34% and 38%) while enhancing the PR interval (7%) and increasing ST height (2.0- and 1.8-fold, respectively). These findings indicate that RVS and PTS contribute to the restoration of cardiac contractility, supporting the clinical relevance of statin therapy in diabetic cardiovascular complications.

A recent study conducted by Liu *et al.* suggests that RVS administration in post-conditioning can reduce the severity of cardiac damage caused by ischemia-reperfusion in isolated hearts [57]. Furthermore, chronic statins treatment protected the integrity of microvasculature in the course of acute myocardial infarction [58], regardless of the presence of lipid-lowering effects. Evidence indicates that early and chronic statin treatment improves myocardial perfusion and reduces infarction regions during ischemia-reperfusion [58–60]. These results align with our findings, which demonstrate the potent anti-inflammatory effects of statins. Treatment with RVS resulted in a 52% reduction in NLRP3 levels and a 28% decrease in IL-1β content, while PTS led to an even greater suppression of NLRP3 levels (62%) and a 24% reduction in IL-1β content relative to the untreated DCM group. These results suggest that RVS and PTS mitigate DCM progression by attenuating key inflammatory pathways, highlighting their potential in reducing inflammation-driven cardiac injury.

Our findings also reveal the differential effects of RVS and PTS on the TLR4/NF-κB, NLRP3 inflammasome, and Akt/GSK3β pathways, underscoring their potential to modulate critical signaling cascades implicated in DCM. This provides novel evidence of statins' cardioprotective effects beyond their lipid-lowering action, particularly in mitigating the inflammatory and fibrotic processes central to DCM pathology [61] attributed the positive impact of RSV on DCM to the inhibition of the NLRP3 inflammasome, which was associated with a reduction in MAPK activation.

The signaling cascade was further validated by examining the intermediary protein RISK pathway involving Akt/GSK-3β, which is closely associated with the adverse effects of hyperglycemia. Additionally, our study elucidates the dual role of the Akt/GSK3β pathway in DCM. While Akt activation is typically protective, chronic activation in the context of hyperglycemia exacerbates cardiac fibrosis and hypertrophy, contributing to long-term cardiac dysfunction. This nuanced understanding of the Akt/GSK3β pathway in DCM progression offers new therapeutic targets for managing the condition. Akt, a key regulator in the PI3K/Akt pathway, promotes cell survival, growth, and metabolism. However, in diabetes, particularly in DCM, chronic hyperglycemia and AGEs upregulate Akt activity, leading to diverse cardiac outcomes [62]. While Akt activation can protect cardiac cells from apoptosis, promote cell survival, and support adaptive responses under stress, persistent and excessive activation can lead to maladaptive effects, including cardiac hypertrophy, fibrosis, and contractile dysfunction, ultimately contributing to cardiac dysfunction and heart failure.

GSK-3β, a downstream target of Akt, is active in its dephosphorylated form, and its heightened activity exacerbates inflammation, fibrosis, and cardiac remodeling in diabetes. These findings underscore the dual role of Akt activation and high-light the therapeutic potential of targeting the Akt/GSK-3β axis in managing diabetic cardiomyopathy [63,64]. The detrimental effects caused by persistent and excessive Akt activation can lead to maladaptive responses, contributing to cardiac hyper-trophy, fibrosis, and contractile dysfunction, which may ultimately result in cardiac dysfunction and heart failure [65,66].

In the F/Fr/STZ model of DCM, oxidative burden was markedly elevated, as evidenced by a 1.7-fold increase in cardiac MDA levels, a key biomarker of lipid peroxidation, accompanied by a significant 22% reduction in GSH content, a critical endogenous antioxidant. These findings underscore the profound oxidative damage associated with DCM. Treatment with RVS and PTS demonstrated substantial antioxidative effects, as both treatments significantly reduced MDA levels in car-diac tissues by 41% and 39%, respectively. Additionally, they restored antioxidant defenses, evidenced by a 28% increase in GSH content. These results highlight the ability of RVS and PTS to mitigate oxidative stress, thereby protecting cardiac tissues from ROS-induced damage and contributing to their therapeutic potential in DCM. Previous studies demonstrated that statins have the ability to boost antioxidant defenses verified by the improvement of GSH and reduction of MDA [67,68]. Another study revealed that after statin treatment of human endothelial cells in vitro, there is an observed increase in the expression of antioxidant enzymes, with the regulation of this process being associated with Akt phosphorylation at Ser473, which in turns reduces the levels of DNA damage [69].

Although the mechanisms through which statins influence the RISK pathway and modulate oxidative stress are still being explored, several key insights have emerged from ongoing research, as illustrated in Fig 14. These findings

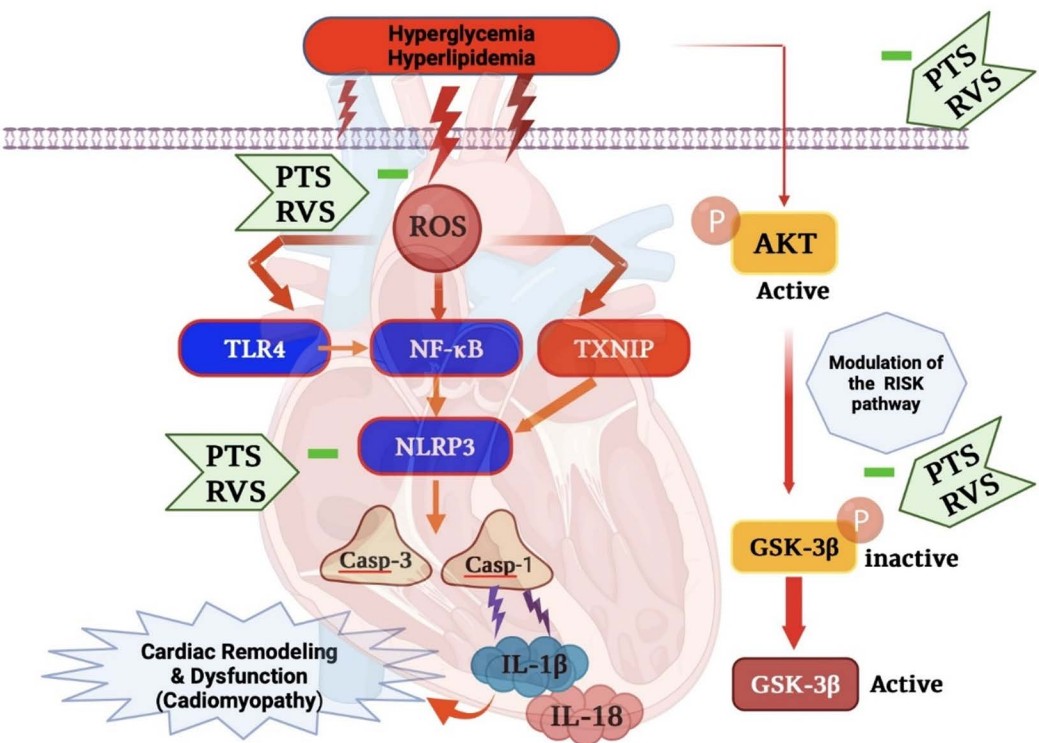

**Fig 14. The proposed protective mechanism of Rosuvastatin and Pitavastatin in mitigating F/Fr/STZ-induced DCM in rats via RISK, NF- κB/ NLRP3 inflammasome and TLR4/ NF-κB signaling pathways.** GSH: reduced glutathione; MDA: malondialdehyde; P-GSK-3 β: phosphorylated-Glycogen synthase kinase-3;P-AKT: phosphorylated- serine/threonine protein kinase; IL-1β: Interleukin-1 beta; NLRP3: NOD-like protein 3; ROS: reactive oxygen species.

highlight the proposed protective mechanism of RVS and PTS in mitigating F/Fr/STZ-induced DCM in rats. Specifically, RVS and PTS are suggested to exert their effects through modulation of multiple critical signaling pathways, including the RISK pathway, NF-κB/NLRP3 inflammasome, and TLR4/NF-κB cascades [18,53,54–70]. The RISK pathway, which is activated by PI3K/Akt signaling, plays a central role in myocardial protection during ischemia-reperfusion, providing cellular survival mechanisms against oxidative stress.

Statins may enhance the cell's ability to withstand reperfusion-associated stress, thereby minimizing damage to heart tissue. Specifically, statins are known to influence proteins and enzymes involved in the RISK cascade, with studies demonstrating that statins modulate key signaling molecules such as PI3K, Akt, and GSK-3β, which are integral to myocardial protection during ischemia-reperfusion. For example, research by Bland et al. (2022) has shown that statins may enhance Akt phosphorylation, activating the RISK pathway and protecting the heart from oxidative stress and reperfusion injury. A previous study also documented that RVS could prevent myocardial ischemia-reperfusion injury by regulating the phosphorylation of PI3K/Akt and GSK-3β, which plays a role in mitigating oxidative stress and subsequent damage to the heart [57]. However, other studies have proposed that statins, including RVS, can induce myotoxicity and impair Akt activity at high concentrations [71]. Furthermore, RVS postconditioning may prevent myocardial ischemia-reperfusion injury by inducing phosphorylation of PI3K/Akt and GSK-3β, while simultaneously promoting a higher $Ca^{2+}$ load required to prevent mPTP opening, thereby mitigating oxidative stress. Additionally, RVS postconditioning significantly increases superoxide dismutase activity and reduces MDA and ROS levels, further protecting against reperfusion injury [62].

Statin therapy has been observed to have a dual impact on inflammation-related factors, particularly IL-1β. On one hand, statins are associated with reduced NLRP3 activation and downstream mediators like IL-1β, which may contribute to improved cardiac function For instance, one study demonstrated that statins could decrease NLRP3 gene expression in peripheral blood mononuclear cells from patients with cardiovascular disease, suggesting their anti-inflammatory potential under certain conditions [72]. This reduction in IL-1β expression is thought to be related to statins' ability to inhibit the activation of pro-inflammatory pathways, including NF-κB and the inflammasome, which are associated with the production of IL-1β during inflammatory responses.

On the other hand, several studies have indicated that statins can augment the expression of IL-1β when lipopolysaccharides (LPS) are present, particularly in the absence of NLRP3 inflammasome activation. For example, a study by Sheridan et al. (2022) demonstrated that statins may enhance IL-1β expression in LPS-stimulated macrophages, suggesting that the inflammatory response to pathogens could be exacerbated under certain conditions [73]. This increase in IL-1β expression has been attributed to statins' potential to activate alternative inflammatory pathways, such as the upregulation of the NF-κB pathway independent of NLRP3 inflammasome activation, or the modulation of TLR4 signaling, both of which could increase IL-1β production in the presence of an immune stimulus. Thus, the effects of statins on IL-1β expression appear to be context-dependent, varying according to the specific inflammatory environment, the presence of pathogens, and the balance between inflammasome activation and NF-κB signaling. These opposing actions highlight the complexity of statins' inflammatory modulation, underscoring the need for further studies to elucidate the precise mechanisms that govern these effects and their implications for cardiovascular therapy".

Moreover, statins, including RVS, modulate key antioxidant enzymes such as thioredoxin-1 and heme oxygenase-1 (HO-1), which play crucial roles in protecting the heart from oxidative stress in DCM. Statins enhance thioredoxin-1 activity by promoting its S-nitrosylation, reducing intracellular ROS and oxidative damage. They also upregulate HO-1 expression through the PI3K/Akt pathway, further alleviating oxidative stress and preventing myocardial fibrosis, hypertrophy, and dysfunction. These pleiotropic effects of statins, beyond their lipid-lowering action, have important therapeutic implications in protecting cardiac tissues from oxidative injury in DCM. Statin boosted production and S-nitrosylation of thioredoxin 1, boosting its activity and lowering intracellular ROS in human endothelial cells with the increased HO-1 mRNA and protein [74]. Statins also protected the antioxidant effects of high-density lipoprotein by increasing paraoxonase-1 (PON1) activity, which was boosted in hypercholesterolemic patients [75]. Oxidative stress can cause abnormal heart rhythms,

as documented in a previous clinical study where statins were tested for their impact on ROS generation in heart surgery patients and those with persistent atrial fibrillation [76]. As discussed earlier, DCM is initiated by the crosstalk of oxidative stress and inflammation, with the activation of TLR4/NF-κB cascades as shown in Fig 14. It has been demonstrated RVS's pleiotropic ability to inhibit the lipopolysaccharides-induced activation of TLR4/NF-κB cascades in myocardial tissue [70]. Likewise, PTS reduced the deterioration in cardiac muscle function in the mice cardiac failure model, which was correlated with its anti-inflammatory effect [77].

Finally, histopathological examination revealed normal myocardial architecture in the control group, with well-organized myofibrils and no signs of inflammation or fibrosis. In contrast, the dbsF/Fr/STZ-induced DCM model showed significant inflammatory cell infiltration, collagen deposition, fat accumulation in the myocardium, indicating fibrosis and structural damage. Treatment with RSV reduced inflammatory infiltration and partially restored myocardial structure, though some irregularities remained. PTS treatment resulted in more normalized myocardial architecture, with well-aligned myocardial fibers and a size closer to the control group, suggesting its effectiveness in mitigating cardiac damage. Both RSV and PTS demonstrated potential in improving myocardial structure and reducing inflammation in DCM. Together, these histo-pathological findings demonstrate the potential of both RSV and PTS in reversing the structural and inflammatory changes associated with DCM, with PTS showing a more normalized myocardial architecture compared to RSV. This fibrosis impairs heart function and contributes to heart failure. Treatment with RSV reduced fibrosis to a moderate degree, while PTS treatment resulted in minimal fibrosis and nearly normal myocardial architecture, with well-aligned fibers and restored cardiac myocyte size. These findings suggest that PTS is particularly effective in reducing fibrosis and preserving heart structure in DCM.

## 5. Conclusion

This study demonstrates the prominent cardioprotective potential of both RSV and PTS in the context of DCM in rats. These statins effectively improved cardiac function, reduced oxidative stress, and mitigated inflammatory pathways, including NLRP3 inflammasome and NF-κB/TLR-4 signaling, as well as the RISK signaling pathway. Molecular signaling imbalances in DCM were also addressed by the statin treatments. Furthermore, histopathological examinations supported these findings by showing a reduction in inflammatory cell infiltration and fibrosis. These results collectively highlight the promising therapeutic role of statins, particularly RSV and PTS, in mitigating the adverse effects of DCM and suggest their potential role for clinical applications in managing this condition.

## 6. Limitations of the study

This study provides valuable insights into the protective effects of RVS and PTS in mitigating F/Fr/STZ-induced DCM, yet several limitations must be acknowledged. One significant limitation is the lack of data on the long-term effects of RVS and PTS in DCM, particularly concerning their safety and efficacy over extended periods. While the current findings highlight promising mechanisms, the translation of these results to clinical practice poses challenges, including differences between animal models and human physiology.

## 7. Future directions

Further research is needed to determine whether these effects can be replicated in clinical settings, especially among patients with varying stages of DCM or comorbidities such as hypertension and obesity. Investigating the potential inter-actions of RVS and PTS with other therapeutic agents and evaluating their impact on broader cardiovascular outcomes would also be essential for validating their therapeutic potential.

While this study provides compelling evidence for the potential of RSV and PTS in mitigating DCM in animal models, translating these findings to human patients requires careful consideration of several factors. In humans, dosing, treat-ment duration, and patient demographics (e.g., age, comorbidities such as diabetes or hypertension) will play crucial roles

in determining the therapeutic efficacy and safety of RSV and PTS. Therefore, future research should focus on designing randomized controlled trials to assess the safety and efficacy of RSV and PTS in diverse human populations, taking into account variations in treatment responses. Moreover, studies should explore the long-term effects of RSV and PTS on cardiac function, fibrosis, and inflammation markers to establish their therapeutic potential in human DCM. Additionally, combining RSV and PTS with other therapeutic agents might yield synergistic effects, warranting further investigation into combination therapies.

## Acknowledgement

The publication of this article was funded by the Open Access Fund of Leibniz University Hannover.

## Supporting information

**S1 File. Supplementary File.** This file includes additional data and materials that support the findings reported in the main manuscript. It contains supplementary tables, figures, and methodological details referenced in the main text. (DOCX)

## Author contributions

**Conceptualization:** Dalia O. Saleh, Nesma M.E. Abo El Nasr, Ingy M. Hashad.

**Data curation:** Dalia O. Saleh, Nesma M.E. Abo El Nasr, Marawan A. Elbaset, Marwa E. Shabana, Ingy M. Hashad.

**Formal analysis:** Dalia O. Saleh, Nesma M.E. Abo El Nasr, Marawan A. Elbaset, Marwa E. Shabana, Ingy M. Hashad.

**Funding acquisition:** Tuba Esatbeyoglu.

**Investigation:** Dalia O. Saleh, Nesma M.E. Abo El Nasr, Marawan A. Elbaset, Marwa E. Shabana, Ingy M. Hashad.

**Methodology:** Dalia O. Saleh, Nesma M.E. Abo El Nasr, Marawan A. Elbaset, Marwa E. Shabana, Ingy M. Hashad.

**Project administration:** Dalia O. Saleh, Nesma M.E. Abo El Nasr, Marawan A. Elbaset, Marwa E. Shabana, Sherif M. Afifi, Ingy M. Hashad.

**Resources:** Dalia O. Saleh, Nesma M.E. Abo El Nasr, Marawan A. Elbaset, Marwa E. Shabana, Sherif M. Afifi, Ingy M. Hashad.

**Software:** Dalia O. Saleh, Nesma M.E. Abo El Nasr, Marawan A. Elbaset, Marwa E. Shabana, Ingy M. Hashad.

**Supervision:** Dalia O. Saleh, Nesma M.E. Abo El Nasr, Ingy M. Hashad.

**Validation:** Dalia O. Saleh, Nesma M.E. Abo El Nasr, Marawan A. Elbaset, Marwa E. Shabana, Ingy M. Hashad.

**Visualization:** Dalia O. Saleh, Nesma M.E. Abo El Nasr, Marawan A. Elbaset, Marwa E. Shabana, Ingy M. Hashad.

**Writing – original draft:** Dalia O. Saleh, Nesma M.E. Abo El Nasr, Marawan A. Elbaset, Marwa E. Shabana, Ingy M. Hashad.

**Writing – review & editing:** Dalia O. Saleh, Nesma M.E. Abo El Nasr, Marawan A. Elbaset, Marwa E. Shabana, Tuba Esatbeyoglu, Sherif M. Afifi, Ingy M. Hashad.

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
