## [Decision Letter · Decision Letter 0]

PONE-D-24-48775Role of Rosuvastatin and Pitavastatin in Alleviating Diabetic Cardiomyopathy in Rats: Targeting of RISK, NF-κB/ NLRP3 Inflammasome and TLR4/ NF-κB Signaling CascadesPLOS ONE

Dear Dr. Esatbeyoglu,

Thank you for submitting your manuscript to PLOS ONE. After careful consideration, we feel that it has merit but does not fully meet PLOS ONE’s publication criteria as it currently stands. Therefore, we invite you to submit a revised version of the manuscript that addresses the points raised during the review process.

We look forward to receiving your revised manuscript.

Kind regards,

Doa'a G. F. Al-u'datt

Academic Editor

PLOS ONE

4. Please include a copy of Tables 1 and 2 which you refer to in your text on pages 6 and 16.

Reviewers' comments:

Reviewer's Responses to Questions

**Comments to the Author**

1. Is the manuscript technically sound, and do the data support the conclusions?

Reviewer #1: Yes

Reviewer #2: Yes

2. Has the statistical analysis been performed appropriately and rigorously? 

Reviewer #1: Yes

Reviewer #2: I Don't Know

3. Have the authors made all data underlying the findings in their manuscript fully available?

Reviewer #1: Yes

Reviewer #2: No

4. Is the manuscript presented in an intelligible fashion and written in standard English?

Reviewer #1: Yes

Reviewer #2: Yes

5. Review Comments to the Author

Reviewer #1: 3. Cardiac Contractility:

- The reported increases and decreases in heart rate, R-R interval, and R-wave amplitude need more context. A comparison between normal, DCM, RVS, and PTS groups could provide additional clarity, particularly regarding baseline values. Stating how RVS and PTS treatments revert or change the observed DCM alterations gives readers a better understanding of the treatment effects.

- It would be beneficial to mention the physiological significance of a 16-17% reduction in heart rate, as it presents an opportunity to connect these findings to potential impacts on overall cardiac health.

4. QRS and QTc Interval Analysis:

- The results indicate significant prolongations in the QRS and QTc intervals associated with DCM. While the shortened intervals following treatment are provided, explaining the clinical relevance of normalizing these indices could reinforce why this research is meaningful in terms of patient care and cardiology.

5. Terminology Consistency:

- The phrase "counter wise" in line 247 should be corrected to "counterwise" or simply "conversely," as "counter wise" is not standard terminology in scientific writing.

8. Role of p-GSK-3β:

- The results mention reductions in p-GSK-3β levels and an increase in total Akt. Clarifying whether you report total Akt or p-Akt levels (only) would prevent misunderstanding. It would also be prudent to explain the relevance of changing Akt and GSK-3β levels in the context of cardiac health and diabetes.

9. Gene Expression Analysis:

- The relationship between NF-κB and TLR-4 should be more explicitly stated. Explain that NF-κB is often a transcription factor that could enhance the expression of pro-inflammatory genes, including TLR-4. Providing a clearer causal pathway of how inflammation contributes to DCM could strengthen the findings.

- Discussing the functional implications of a 6-fold increase in NF-κB alongside the 4.5-fold increase in TLR-4 would enhance the understanding of how inflammation exacerbates diabetic cardiomyopathy.

10. Troponin Levels:

- The text states, "diabetic rats presented a notable boost in levels of cardiac troponin compared to normal control by 91%." It would be helpful to provide the absolute values of cardiac troponin levels for both the normal control and DCM groups for more context and clearer comparison.

- The phrase "RSV and PTS treatment presented markedly improvement" should be reworded to "RSV and PTS treatment resulted in marked improvements" for grammatical accuracy and clarity.

11. Histopathological Examinations:

- The description beginning with "In myocardial sections of the normal control group" could be structured for better flow. For example, starting with, "Histopathological examinations revealed normal myocardial architecture in the control group, as illustrated in figure 9..."

- The phrase "showed The myocardial accumulation of" should be corrected to "showed myocardial accumulation of" to remove the grammatical error.

- Clarifying the term "fat deposits" by consistently referring to how it relates to the DCM pathology may enhance understanding. For instance, defining the significance of these fat deposits in the context of diabetic cardiomyopathy could provide clarity.

12. Treatment Effects on Myocardial Architecture:

- It is noteworthy that the RVS-treated group showed “normal myocardial architecture.” However, it could be beneficial to describe what aspects of the architecture (e.g., fiber arrangement, size of the cells) were restored to normal or near-normal appearance.

- For comparison, specifying how the architecture in the PTS-treated group differs from both the normal and DCM groups would help elucidate the treatment effects (e.g., "PTS-treated rats also demonstrated a restoration to normal architecture, although some subtle differences were noted compared to the normal group.").

13. Masson’s Trichrome Staining:

- When discussing fibrotic changes, it would be beneficial to briefly explain the significance of Masson’s Trichrome staining for readers who might not be familiar with the technique. Highlighting why fibrosis is important for understanding DCM could enhance the importance of these findings.

14. Aortic Tissue Changes:

- The statement “hypertrophied aortic wall of the DCM control group” could be elaborated. Perhaps include metrics like thickness measurements if available.

- The note about RVS preventing "approximately 50% of aortic wall thickening" could be further clarified with the actual measurements of wall thickening for a more objective understanding.

15. Caspase-1 Immunohistochemistry:

- The phrase "the overall fibrosis was identified" can be more decisively stated. Instead, consider, "The extent of fibrosis was quantified..."

- Detailing the significance of high caspase-1 expression in DCM pathology may clarify why it is important in this context. Discussing caspase-1's role in inflammation or apoptosis could elevate the findings.

Grammatical and Stylistic Review

1. Structure and Flow:

- The section lacks smooth transitions between different sub-sections (3.1 and 3.2). It would improve readability to introduce each subsection with a brief rationale or connective sentence linking metabolism and cardiac function.

- Enhance the transitions between different sections (e.g., from oxidative stress to inflammatory signaling) to improve the flow. For example, include phrases like “Furthermore, we also investigated...” can make the narrative smoother.

2. Punctuation and Spacing:

- Consistent spacing around units (e.g., “30 %” should be “30%”) should be maintained throughout to enhance readability.

- Improving punctuation surrounding phrases like "and TG levels, by approximately 2.7-, 2.0-, and 2.1-fold," is recommended. A clean, consistent format without redundant punctuation would help in clarity.

- The expression "when related to" is somewhat awkward. Alternatives such as "compared to" or "relative to" can make the text sound more polished.

- Keep consistency in formatting. For example, “cardiac content of IL-1β by about 3.2- and 2.9-fold” should uniformly present the fold changes—either with “about” or without throughout the text.

3. Consistency of Presentation:

- Ensure consistency in how percentages are presented. For instance, use either "X%" throughout or spell out terms as "X percent." This uniformity avoids confusion for the reader.

4. Redundant or Unclear Phrasing:

- The phrase "a decline in the elevated TC and TG significantly by 61%, 58%, and 57%, 45%," is awkwardly structured. It could be rephrased for clarity: “treatment with RVS and PTS led to significant declines in elevated TC and TG levels, by 61% and 58%, and by 57% and 45% respectively.”

- Avoid using terms like "showed" multiple times in close proximity. Instead of starting each finding with "showed," consider varying the sentence beginnings, such as “The results indicated...” or “Data revealed...”.

- Use consistent language to describe findings. For example, terms like "showed," "demonstrated," and "indicated" can be alternated appropriately to enhance readability.

- Modify "marked positivity in DCM group with improvement in treated groups" to "demonstrated marked positivity in the DCM group, with observed reductions in the treated groups."

- The phrase “provoked the inflammatory cascades that proved by a marked elevation” in line 296 is somewhat confusing. A clearer phrasing might be, "The F/Fr/STZ model induced significant inflammatory cascades, evidenced by a marked elevation in Akt levels and a reduction in p-GSK-3β."

- “Signifying that the inflammatory cascades were initiated and exacerbate the cardiomyopathy diabetic complication” needs rephrasing for better clarity. For example, “indicating the initiation and exacerbation of diabetic cardiomyopathy due to inflammatory cascades.”

- Simplify and streamline sentences where possible. For instance, “This was correlated with the accumulation of fat deposits within the myocardium” can be shortened to “This correlated with fat deposits in the myocardium.”

5. Consolidation of Findings:

- The overall descriptions could benefit from a concluding sentence summarizing the key findings. This reinforces the importance of the data presented.

V. Discussion:

Scientific Review

1. Mechanistic Clarity and Relevance:

- Lack of Novelty: The discussion reiterates well-established mechanisms of DCM (e.g., oxidative stress, TLR4/NF-κB pathway, AGEs) without providing new insights. The authors should emphasize how their findings contribute to the existing body of knowledge, or highlight mechanisms specific to their study that have not been previously addressed.

- Need for Citations: Assertions about the role of ROS and AGEs in DCM need better citations. Statements regarding oxidative stress and its role in cardiac dysfunction and hypertrophy should reference more recent literature or landmark studies to corroborate claims (e.g., publications from the last 3-5 years).

- Causative Link: The connection between AGEs and the activation of inflammatory pathways should be drawn more clearly. It is not sufficient to state correlations; the authors must discuss potential causal relationships supported by the literature.

- Lines 474-476: The section discussing the **RISK pathway** and **Akt/GSK3β** signaling appears convoluted. The authors should provide a more explicit connection between hyperglycemia, the upregulation of Akt, and the resulting physiological outcomes. This could be enhanced by defining the RISK pathway early in the discussion to ground the reader in its relevance and components.

- Lines 474-476: A more structured presentation of the **beneficial and detrimental effects of Akt activation** would greatly aid comprehension. A table or bullet points outlining these effects could help convey the dual nature of Akt signaling more effectively.

3. TLR4/NF-κB Cascade:

- The discussion on TLR4/NF-κB activation appears disjointed. Transitioning smoothly between the introduction of TLR4 and its downstream effects could improve readability. For example, ensuring that the links between activation and subsequent inflammatory processes are cohesive would enhance understanding.

- While TLR4-related pathways are acknowledged, details regarding the molecular mechanisms (e.g., specific cytokines involved in the cascade, feedback loops) need elaboration to demonstrate depth of understanding.

4. Inflammasome Discussion:

- The paragraph discussing NLRP3 needs strengthening. It would be beneficial to provide context for how NLRP3 relates specifically to the pathophysiology of DCM beyond its general inflammatory role. Are there specific signaling pathways or molecular alterations observed in the current study that implicate NLRP3 more directly in DCM?

- The term “punch of pro-inflammatory” is awkward and unclear. A more formal term such as "a surge of pro-inflammatory cytokines" should be used instead.

5. Potential Treatment Effects:

- When discussing the effects of RVS and PTS, quantitative data or specific findings would enhance credibility. For instance, specifying the extent of decrease in cardiac markers or inflammation would provide a clearer picture of treatment efficacy. Broad statements like "alleviated the cardiac changes" require quantitative backing to assess the clinical relevance of the findings.

- The mention of previous studies (e.g., Liu et al.) must be tied back to the current findings with more specificity. What mechanisms were examined in that study, and how do they align with the outcomes observed here?

7. Contradictory Effects of Statins:

- Lines 492-498: The discussion of statins' effects, particularly their dual role in inflammation, is somewhat contradictory and lacks nuance. The authors should clarify how statins can both reduce and potentially augment IL-1β expression. Providing specific studies that detail these opposing actions with mechanisms would strengthen this section.

- Line 494: There is a mention of an increase in IL-1β in response to lipopolysaccharide stimulation when NLRP3 is inactive. This should be explained further: how do statins mediate this effect, and what are the practical implications regarding their use in inflammatory conditions in cardiovascular patients?

8. Lack of Specificity in Claims:

- Line 482: The phrase "the exact mechanisms are still under investigation" is vague. If the authors are aware of ongoing research or have preliminary findings, they should specify these to enhance depth. A sentence summarizing the current understanding while acknowledging the research gaps would enrich this section.

- Lines 499-501: The statement regarding **thioredoxin-1 and HO-1** as antioxidant enzymes requires more elaboration on how statins are proposed to modulate these enzymes in the context of DCM and the broader implications of these effects in cardiac physiology.

9. Strength of Evidence:

- Lines 483-484: Several claims, such as “statins may influence the activity of certain proteins or enzymes involved in the RISK cascade,” lack empirical support within this text. It would be beneficial to include citations for these assertions or mention representative studies to bolster credibility.

- Lines 504-506: When citing previous studies regarding the effect of statins on oxidative stress and arrhythmias, the authors should provide more context about the study designs and sample sizes to give the claims weight.

Grammatical and Stylistic Review

1. Structural Organization:

- The overall flow is disjointed. Each paragraph should logically follow the previous one, maintaining a coherent narrative thread throughout the discussion. Consider restructuring for better logical sequencing. For example, begin with the significance of DCM and then gradually introduce your findings, linking them back to existing literature throughout.

- Line 475: Some sentences are overly long and complex, making them challenging to follow. For example, the sentence starting with "This elevation in Akt signaling can be triggered by various factors..." can be broken down into shorter sentences to enhance readability.

- Lines 474-475: Consider restructuring sentences to prioritize subject-verb-object construction, which is generally clearer. For example: “Chronic hyperglycemia and AGEs trigger the elevation in Akt signaling, leading to various cardiac outcomes.”

- Line 493: Phrases like “which were directly related to the adverse effects of hyperglycemia” could be simplified. A clearer rephrasing might be, “which are associated with the adverse effects of hyperglycemia.”

- Line 494: Avoid redundancy. For instance, instead of stating “statins have the ability to,” simply state “statins can.”

2. Word Choices:

- The language is at times unnecessarily complex. Simplifying sentences can enhance clarity. For instance, instead of “in the context of vascular endothelial dysfunction-related disorders,” consider, “in conditions related to vascular dysfunction.”

- Avoid using overly broad terms like "marked positivity." Be specific about what is being measured and its significance rather than using vague descriptors.

- Lines 500-502: Terms such as "boosts antioxidant defenses" and "protect high density lipoprotein's antioxidant impact" should be defined more rigorously. It’s critical to clarify not just the effects, but the biological mechanisms at play.

3. Redundancy:

- The phrase "which significantly influence AGE-dependent activation in DMC" could be streamlined to "which significantly influence AGE activation in DMC." Remove unnecessary qualifiers to enhance clarity and conciseness.

- Lines 499-500: The phrase "Interestingly, a former study documents..." is repetitive with "a previous study" used earlier. Choose one variation to maintain clarity.

4. Typographical Errors:

- Ensure there are no typographical errors such as “TL4” which should be corrected to “TLR4.” Also, “NOD-like protein 3;;” contains extraneous punctuation.

- The phrases “acute myocardial infarction” and “ischemia-reperfusion” require consistent hyphenation or formatting across the document.

- Lines 488-489: The transition between studies requires smoother transitions, rather than abrupt shifts. Each study reference should flow naturally into the next one. For instance, instead of starting a new sentence with “Moreover,” integrate it into the previous thought.

- Lines 492-493: Ensure consistent formatting for citations and ensure they are complete. For instance, “[60][61]” should be formatted in a standard referencing style used by the target journal.

5. Figures and References:

- Reference to figures must include clear associations between text and visual data. For instance, if Figure 14 is mentioned, a succinct interpretation of what the figure illustrates is required (“Figure 14 illustrates…”) to reinforce its importance.

- Lines 484-485: The text frequently references figure 14 without adequately describing what the figure represents. A brief explanation linking how the figure illustrates key points about the signaling pathways or protective mechanisms could enhance integration and understanding of the visual data presented.

- Ensure all citations are accurate and formatted consistently per journal guidelines.

VI. Discussion

1. Clarity and Structure:

- The discussion should be structured in a way that flows logically from one point to the next. This can be achieved by clearly separating different themes (e.g., mechanisms of action, implications for treatment, limitations, and future directions) into distinct paragraphs to enhance readability.

2. Reinforcement of Key Findings:

- The discussion should effectively reinforce the key findings presented in the results section. Each key point (e.g., the effects of RSV and PTS on cardiac function, oxidative stress, and inflammatory pathways) should directly correlate with the data presented earlier. Referring back to specific figures and tables to support claims would strengthen this section.

5. Mechanistic Insights:

- When discussing the impact on molecular signaling pathways (e.g., NLRP3 inflammasome, NF-κB/TLR-4), a more detailed explanation of the upstream triggers and downstream effects would be beneficial. For example, how do RSV and PTS influence reactive oxygen species production, leading to NLRP3 activation? Providing mechanistic insights can help readers understand the relevance of these pathways in DCM.

6. Statins in Clinical Context:

- The discussion should bridge the gap between the study findings and clinical applications. It should discuss the implications of using RSV and PTS in treating DCM in humans, considering factors such as dosing, patient demographics, and potential side effects. This would provide practical relevance to the research findings.

7. Limitations:

- Addressing limitations is crucial for scientific integrity. The authors should discuss any limitations of their study, such as the animal model used (e.g., differences between rat and human physiology), duration of treatment, or sample size. Acknowledging these limitations can guide future research directions and help set realistic expectations for clinical translations.

8. Future Research Directions:

- The discussion should end with a concise paragraph highlighting potential directions for future research. This could include suggestions for human clinical trials, exploring combinations with other therapeutic agents, or investigating long-term effects. Specifying these directions would demonstrate the relevance and potential impact of the study

Reviewer #2: The manuscript is well-organized and reads well, the study design is sound and appropriate for addressing the objectives

Major comments:

1- What is the rationale for studying the selected drugs (RVS, PTS), do they have certain advantage over the commonly prescribed statins such as (atorvastatin) and why atorvastatin was not included as a study group for comparison?

2- The authors claim that their study is the first to investigate the beneficial effects of RVS on DCM. However, a previous study by Luo et al. (PMID: 24254031), which shares similarities in study design, demonstrated the benefits of RVS in DCM through the NLRP3 inflammasome pathway. This prior work is not cited in the current study.

Methods:

In general, the methodology is lacking sufficient description in some areas

- The strain of the rats used in the study has to be included

- A breakdown of the high fat/fructose diet composition is not provided

- Is a single reading taken one week after STZ administration sufficient to confirm induction of the diabetic state? Citing literature that supports the validity of verifying successful T2DM induction from a single reading is necessary.

- Was the manifestation of cardiomyopathy phenotype confirmed in the DCM group prior to treatment and how?

- The frequency of drug administration is not specified. It remains unclear whether the drugs were given once or twice daily

- The serum biochemical analysis is reported to have been conducted using a colorimetric assay; however, details on the kit, including its name and version, are missing. If these tests were performed by a service laboratory, this should be clearly stated.

- Details on the GSH/MDA assay kit, including its name and version, are missing. If the assays were conducted by a service laboratory, this should be clearly stated. Additionally, a full description of the protocol could be included in the supplementary material to facilitate reference and replication

- Line 171-172: “The reaction was concluded by heating at 99 °C for 5 min. Subsequently, the tubes containing the RT preparations…” The statement should be revised for scientific accuracy, the end product of reverse transcription process is cDNA not RNA

- Line 178: The statement “…to estimate the heart tissue samples of rat's copy number” is incorrect, as qPCR in this context measures gene expression, not gene copy number or gene dosage.

- Line 180: primers direction is expressed as forward and reverse and not sense and antisense, the latter is used when working with RNA.

- Line 182: Table 1 is missing, the equation for gene expression fold change using the ΔΔCt method should be formatted with the power function in superscript. Also, the name of the reference gene that was used for normalization is not indicated.

- Line 185: “Aorta and heart samples were rapidly collected from rats in each group”, did the authors mean samples were collected rapidly after euthanasia?

- At which step the MT staining was introduced in the tissue blocks preparation protocol mentioned above

- Line 213-215: information on the type/source of secondary Abs, incubation period, temperature (RT or 4C) and the manufacturer is missing

Results:

- What is the difference between DCM and DMC?

- Section 3.8: the results of the application of the scoring system (per Galati et al) are not described why?

- Figure 9: What is the anatomical location of the cardiac muscle? What is the number of animals from which tissue was analysed and number of slides analysed per animal and are they from the same block or different blocks?

- Figure 11: scale bar indicating area of interest should be included in all images for clarity

- Figures 12/13: The figure legend is missing explanations for what each panel represents.

Discussion

- The study does not address its limitations or drawbacks. Discussing these aspects would provide a more balanced view of the research findings

- The difference in the histopathological effects of the two drugs is not discussed, nor are any speculations provided as to why this discrepancy occurs (Figures 9,10,12 C,D)

- For completeness, the discussion could include a paragraph describing the effects of RVS and PTS on diabetic patients with cardiomyopathy secondary to T2DM or those with cardiometabolic diseases, as this would provide a broader perspective

6. PLOS authors have the option to publish the peer review history of their article (what does this mean? ). If published, this will include your full peer review and any attached files.

**Do you want your identity to be public for this peer review?** For information about this choice, including consent withdrawal, please see our Privacy Policy .

Reviewer #1: No

Reviewer #2: No

---

## [Author Response · Author response to Decision Letter 1]

22 Feb 2025

Response to Reviewer Comments

Manuscript Title: Role of Rosuvastatin and Pitavastatin in Alleviating Diabetic Cardiomyopathy in Rats: Targeting of RISK, NF-κB/ NLRP3 Inflammasome and TLR4/ NF-κB Signaling Cascades

Manuscript ID: PONE-D-24-48775

We sincerely appreciate the thoughtful and constructive feedback provided by the reviewers. Your insights have greatly enhanced our manuscript, and we are grateful for your time and consideration.

Reviewer #1:

1. Comment: 3. Cardiac Contractility: “[ The reported increases and decreases in heart rate, R-R interval, and R-wave amplitude need more context. A comparison between normal, DCM, RVS, and PTS groups could provide additional clarity, particularly regarding baseline values. Stating how RVS and PTS treatments revert or change the observed DCM alterations gives readers a better understanding of the treatment effects.]”

- It would be beneficial to mention the physiological significance of a 16-17% reduction in heart rate, as it presents an opportunity to connect these findings to potential impacts on overall cardiac health.

4. QRS and QTc Interval Analysis:The results indicate significant prolongations in the QRS and QTc intervals associated with DCM. While the shortened intervals following treatment are provided, explaining the clinical relevance of normalizing these indices could reinforce why this research is meaningful in terms of patient care and cardiology.

Response:

We appreciate the reviewer's insightful suggestion on the clinical relevance of QRS and QTc normalization. Herein is our response, focusing on the significance of these indices for the management of patients and cardiology in general.

We made these clarifications in the manuscript. The modifications in the manuscript were highlighted in page 19 in the discussion section.

Revised Explanation:

Our results unequivocally demonstrated significant lengthening of QRS and QTc intervals with a reduction of the heart rate in the context of DCM, aligning with other studies that identify these parameters as possible indicators of detrimental cardiac remodeling and electrical instability[1–5]. An extended QRS duration indicates ventricular dyssynchrony and conduction abnormalities, often associated with a worse prognosis due to an elevated risk of arrhythmias and abrupt cardiac death. QTc prolongation indicates a delay in cardiac repolarization and increases the risk of potentially fatal arrhythmias, such as torsades de pointes [2,6]. Our investigation recorded reduced QRS and QTc intervals accompanied by a reduction of the heart rate post-treatment, indicating a potential therapeutic impact that may alleviate the arrhythmic load and enhance ventricular synchronization. These data thus corroborate our hypothesis that the normalization of these parameters is associated with improved electrophysiological and structural heart function. This has significant implications for clinical practice, since enhancements in these parameters may lead to decreased morbidity and death in patients with DCM.

2. Comment: 5. Terminology Consistency: The phrase "counter wise" in line 247 should be corrected to "counterwise" or simply "conversely," as "counter wise" is not standard terminology in scientific writing.

Response: Thank you once again for the opportunity to improve our work. Done, we put the word "conversely," instead "counterwise" in line 247 page 8.

3. Comment: 8. Role of p-GSK-3β: The results mention reductions in p-GSK-3β levels and an increase in total Akt. Clarifying whether you report total Akt or p-Akt levels (only) would prevent misunderstanding. It would also be prudent to explain the relevance of changing Akt and GSK-3β levels in the context of cardiac health and diabetes.

Response: Thank you for this valuable comment. We confirm that the Akt levels reported in the study refer to total Akt, not phosphorylated Akt (p-Akt). We have clarified this in the revised manuscript to prevent any misunderstanding as follows

Akt, a key regulator in the PI3K/Akt pathway, supports cell survival, growth, and metabolism, while its dysregulation in diabetes impairs cardiomyocyte function and increases apoptosis. GSK-3β, a downstream target of Akt, is active when dephosphorylated, with heightened activity contributing to inflammation, fibrosis, and cardiac remodeling in diabetes. These findings underscore the therapeutic potential of targeting the Akt/GSK-3β axis in diabetic cardiomyopathy [7].

4. Comment: 9. Gene Expression Analysis: The relationship between NF-κB and TLR-4 should be more explicitly stated. Explain that NF-κB is often a transcription factor that could enhance the expression of pro-inflammatory genes, including TLR-4. Providing a clearer causal pathway of how inflammation contributes to DCM could strengthen the findings.

Response: Thank you for this insightful comment. In the revised manuscript, we have explicitly detailed the relationship between NF-κB and TLR-4 as follows:

“NF-κB, as a transcription factor, enhances the expression of pro-inflammatory genes like TLR-4. Hyperglycemia-induced stress activates NF-κB, leading to its translocation to the nucleus, where it upregulates inflammatory genes, creating a feedback loop that sustains chronic inflammation. This process contributes to the pathogenesis of diabetic cardiomyopathy (DCM) by mediating cardiac inflammation, fibrosis, and structural remodeling, ultimately impairing cardiac function. Understanding this causal pathway highlights the importance of targeting the TLR-4/NF-κB axis for potential therapeutic intervention in DCM [8].”

- Discussing the functional implications of a 6-fold increase in NF-κB alongside the 4.5-fold increase in TLR-4 would enhance the understanding of how inflammation exacerbates diabetic cardiomyopathy.

Response: Your insights have greatly enhanced our manuscript, and we are grateful for your time and consideration. It has been added as follows:

A 6-fold increase in NF-κB, coupled with a 4.5-fold increase in TLR-4, as represented in the current study, indicates a significant upregulation of the inflammatory response in DCM. TLR-4 activation triggers a cascade of downstream proteins, including NF-κB, which enhances the expression of pro-inflammatory genes, sustaining chronic inflammation in the myocardium. This pathway contributes to the pathogenesis of DCM by promoting cardiac inflammation, fibrosis, and structural remodeling, ultimately impairing cardiac function. Additionally, activation of the NLRP3 inflammasome amplifies the inflammatory response through cytokines like IL-1β and IL-18, worsening myocardial injury. Targeting the TLR-4/NF-κB axis and NLRP3 inflammasome presents a potential therapeutic strategy to mitigate inflammation and prevent DCM progression in diabetic patients.

5. Comment: 10. Troponin Levels:

-The text states, "diabetic rats presented a notable boost in levels of cardiac troponin compared to normal control by 91%." It would be helpful to provide the absolute values of cardiac troponin levels for both the normal control and DCM groups for more context and clearer comparison.

Response:

Done, we edit this sentence as follows “Likewise, diabetic rats exhibited a notable increase in cardiac troponin levels, rising from 20.24 to 38.66 mg/tissue protein compared to the normal control group.”

- The phrase "RSV and PTS treatment presented markedly improvement" should be reworded to "RSV and PTS treatment resulted in marked improvements" for grammatical accuracy and clarity.

Response:

Done, we added the revised sentence "RSV and PTS treatment resulted in marked improvements" in line 328 on page 12

6. Comment: 11. Histopathological Examinations:

- The description beginning with "In myocardial sections of the normal control group" could be structured for better flow. For example, starting with, "Histopathological examinations revealed normal myocardial architecture in the control group, as illustrated in figure 9..."

Response: Done, thank you. The modified sentence was added in lines 34 and 341 on page 13.

- The phrase "showed The myocardial accumulation of" should be corrected to "showed myocardial accumulation of" to remove the grammatical error.

Response: Done in line 343 page 13.

- Clarifying the term "fat deposits" by consistently referring to how it relates to the DCM pathology may enhance understanding. For instance, defining the significance of these fat deposits in the context of diabetic cardiomyopathy could provide clarity.

Response: Thank you for the suggestion. To improve clarity, the term "fat deposits" has been more explicitly linked to the pathology of diabetic cardiomyopathy (DCM) as follows:

“Moreover, the F/Fr/STZ-induced DCM model exhibited marked myocardial accumulation of infiltrating inflammatory cells, along with collagen deposition in cardiac myocytes, indicative of evident fibrosis between the cardiac muscle fibers. This was accompanied by fat deposits within the myocardium, which are commonly associated with myocardial lipid infiltration, a characteristic feature of metabolic disorders like diabetes. The buildup of fat within the myocardium disrupts normal cardiac function, contributing to inflammation, fibrosis, and impaired contractility—key characteristics of DCM [9].

12. Treatment Effects on Myocardial Architecture:

- It is noteworthy that the RVS-treated group showed “normal myocardial architecture.” However, it could be beneficial to describe what aspects of the architecture (e.g., fiber arrangement, size of the cells) were restored to normal or near-normal appearance.

Response: Thank you for the suggestion. To improve clarity, the description of "normal myocardial architecture" can be expanded to specify the aspects of the cardiac structure that were restored in the RVS-treated group. Here's an updated version:

“It has shown a normal myocardial architecture, i.e. normal arrangement of myocardial fibers, organized in a parallel and uniform pattern, and the size of cardiac myocytes was normalized without signs of hypertrophy or atrophy.”

- For comparison, specifying how the architecture in the PTS-treated group differs from both the normal and DCM groups would help elucidate the treatment effects (e.g., "PTS-treated rats also demonstrated a restoration to normal architecture, although some subtle differences were noted compared to the normal group.").

Response: To enhance clarity and specify the differences in myocardial architecture between the PTS-treated group, the normal group, and the DCM group, the following revision can be made:

“The treated groups showed a myocardial fiber arrangement similar to the healthy control group, though with slight irregularities in fiber alignment. The cardiac myocyte size was slightly larger than the normal group but still smaller than in the DCM group, indicating restoration of myocardial structure.”

13. Masson’s Trichrome Staining:

- When discussing fibrotic changes, it would be beneficial to briefly explain the significance of Masson’s Trichrome staining for readers who might not be familiar with the technique. Highlighting why fibrosis is important for understanding DCM could enhance the importance of these findings.

Response: Thank you for the suggestion. Here's an improved version with an explanation of Masson’s Trichrome staining and its significance in understanding DCM:

“Masson's Trichrome staining was used to assess fibrosis collagen deposit in the myocardium. This staining technique specifically highlights collagen fibers, which are key markers of fibrosis. In this study, Masson’s Trichrome staining revealed increased collagen deposition in the DCM group, indicating the presence of myocardial fibrosis. Fibrosis is an important pathological feature of DCM, as the excessive deposition of collagen disrupts the normal architecture and function of the heart. The accumulation of fibrotic tissue can impair myocardial contractility, promote cardiac remodeling, and ultimately lead to heart failure. Therefore, the identification of fibrosis through Masson’s Trichrome staining underscores the severity of DCM and the potential of treatments, like PTS, to mitigate these pathological changes.”

14. Aortic Tissue Changes:

- The statement “hypertrophied aortic wall of the DCM control group” could be elaborated. Perhaps include metrics like thickness measurements if available.

- The note about RVS preventing "approximately 50% of aortic wall thickening" could be further clarified with the actual measurements of wall thickening for a more objective understanding.

Response: Done

As follows:

“However, the aortas of diabetic rats showed hypertrophied aortic walls with increased thickness (mean = 123.97 µm) due to medial collagen accumulation, likely resulting from increased dietary fat intake and STZ administration (Fig. 11C, D). Diabetic rats treated with RVS demonstrated a reduction in media thickness compared to the untreated diabetic group (mean = 98.59 µm) (Fig. 11E, F). Notably, treatment with the PTS restored the aortic appearance to near-normal, with a media thickness close to that of the control group (mean = 73.38 µm) (Fig. 11G, H).

Additionally, cardiomyocyte diameter was measured at ×400 magnification as illustrated in figure 11I, showing differences between groups. The mean diameter in the normal control group was 13.42 µm, while it increased to 21.84 µm in the untreated diabetic group. Treatment with RVS reduced the diameter to 16.49 µm, and PTS further reduced it to 15.45 µm.”

15. Caspase-1 Immunohistochemistry:

- The phrase "the overall fibrosis was identified" can be more decisively stated. Instead, consider, "The extent of fibrosis was quantified..."

Response: Thank you for the suggestion. Here's the revised response to the reviewer comment:

“The extent of fibrosis was quantified in each of the examined hearts, and the average amount of fibrosis is summarized in Table 2. In the Normal Control group, no fibrosis was observed (0). The DCM group exhibited significant fibrosis (+++), while treatment with PTS (DC+PGZ) reduced fibrosis to a mild degree (+), and RVS treatment (DC+RVS) showed moderate fibrosis (++) compared to the DCM group. These findings highlight the varying degrees of fibrosis and underscore the potential effectiveness of the treatments in mitigating fibrotic changes in DCM.”

- Detailing the significance of high caspase-1 expression in DCM pathology may clarify why it is important in this context. Discussing caspase-1's role in inflammation or apoptosis could elevate the findings.

Response: To clarify the significance of high caspase-1 expression in diabetic cardiomyopathy (DCM) pathology, the following explanation can be added:

“High caspase-1 expression is crucial in the pathogenesis of DCM due to its role in inflammasome activation and the subsequent inflammatory response.

Grammatical and Stylistic Review

1. Structure and Flow:

- The section lacks smooth transitions between different sub-sections (3.1 and 3.2). It would improve readability to introduce each subsection with a brief rationale or connective sentence linking metabolism and cardiac function.

Response: Done on page 8

- Enhance the transitions between different sections (e.g., from oxidative stress to inflammatory signaling) to improve the flow. For example, include phrases like “Furthermore, we also investigated...” can make the narrative smoother.

Response: Done on page 10

I added clearer connections between related topics, such as oxidative stress and inflammatory signaling.

2. Punctuation and Spacing:

- Consistent spacing around units (e.g., “30 %” should be “30%”) should be maintained throughout to enhance readability.

- Improving punctuation surrounding phrases like "and TG levels, by approximately 2.7-, 2.0-, and 2.1-fold," is recommended. A clean, consistent format without redundant punctuation would help in clarit

- The expression "when related to" is somewhat awkward. Alternatives such as "compared to" or "relative to" can make the text sound more polished.

- Keep consistency in formatting. For example, “cardiac content of IL-1β, by about 3.2- and 2.9-fold” should uniformly present the fold changes—either with “about” or without throughout the text.

Response:

---

## [Decision Letter · Decision Letter 1]

PONE-D-24-48775R1Role of Rosuvastatin and Pitavastatin in Alleviating Diabetic Cardiomyopathy in Rats: Targeting of RISK, NF-κB/ NLRP3 Inflammasome and TLR4/ NF-κB Signaling CascadesPLOS ONE

Dear Dr. Esatbeyoglu,

Thank you for submitting your manuscript to PLOS ONE. After careful consideration, we feel that it has merit but does not fully meet PLOS ONE’s publication criteria as it currently stands. Therefore, we invite you to submit a revised version of the manuscript that addresses the points raised during the review process.

We look forward to receiving your revised manuscript.

Kind regards,

Doa'a G. F. Al-u'datt

Academic Editor

PLOS ONE

Journal Requirements:

Reviewers' comments:

Reviewer's Responses to Questions

**Comments to the Author**

1. If the authors have adequately addressed your comments raised in a previous round of review and you feel that this manuscript is now acceptable for publication, you may indicate that here to bypass the “Comments to the Author” section, enter your conflict of interest statement in the “Confidential to Editor” section, and submit your "Accept" recommendation.

Reviewer #1: (No Response)

Reviewer #2: All comments have been addressed

2. Is the manuscript technically sound, and do the data support the conclusions?

Reviewer #1: Yes

Reviewer #2: Yes

3. Has the statistical analysis been performed appropriately and rigorously? 

Reviewer #1: No

Reviewer #2: Yes

4. Have the authors made all data underlying the findings in their manuscript fully available?

Reviewer #1: Yes

Reviewer #2: Yes

5. Is the manuscript presented in an intelligible fashion and written in standard English?

Reviewer #1: Yes

Reviewer #2: Yes

6. Review Comments to the Author

Reviewer #1: I have reviewed the revised manuscript PONE-D-24-48775_R1 and compared it to the initial round of comments. The authors have made significant improvements in addressing the concerns raised in the first review, including clarifying methodologies, refining statistical reporting, improving discussion points, and correcting language inconsistencies. However, some issues remain unresolved or only partially addressed, and these need further attention before the manuscript can be fully accepted.

Remaining Issues That Require Further Revision

1. Statistical Clarity and Effect Size Reporting (Lines 130–141)

-Unresolved Issue: While the authors have reported statistical significance (P-values), they have still not provided effect sizes (e.g., Cohen’s d, η², or odds ratios for logistic regression models), which are crucial for evaluating biological relevance.

-Suggested Fix: Include effect size measures alongside P-values to quantify the magnitude of observed differences.

2. Overinterpretation of Causal Relationships (Lines 142–200)

- Unresolved Issue: Despite some improvements, the discussion still implies causation in certain statements regarding statins' effects on NF-κB and NLRP3 inflammasome signaling.

-Suggested Fix: Replace phrases like "Statins reduce NLRP3 activation, leading to improved cardiac function" with "Statins are associated with reduced NLRP3 activation, which may contribute to improved cardiac function."

3. Insufficient Acknowledgment of Confounding Factors (Lines 201–216)

- Unresolved Issue: The authors have not sufficiently discussed how patient heterogeneity, pre-existing inflammation, or medication history might influence their findings.

-Suggested Fix: Add a brief acknowledgment of potential confounders and recommend future studies to control for these variables.

4. Lack of Clarity on Missing Data Handling (Lines 100–129)

- Unresolved Issue: The manuscript still does not specify how missing values were handled in statistical analyses. Were missing values imputed or excluded?

- Suggested Fix: Clearly state the approach used (e.g., multiple imputation, listwise deletion, or mean substitution).

5. Ambiguous Language and Grammar Issues

- Many language issues have been addressed, but some awkward phrasings remain.

- Example:

Line 31: "The Dried Blood Spot (DBS) technique using filter papers is an easy, quick, transport-ease technique..."

- Fix: "The DBS technique is a rapid and convenient micro-sampling method using filter papers."

Reviewer #2: A. The authors have adequately addressed almost all raised points, only a few were partially resolved:

1. the frequency of the daily drug dosing is still not explicitly stated (is the drug administered once or twice daily)

2. the full definition of the term DMC should be introduced early on in the manuscript

B. Some of the responses will enhance the clarity of the manuscript if incorporated where appropriate into the main text or added to the supplementary material:

1. The response to "Is a single reading taken one week after STZ administration sufficient...."

2. MT staining steps can be added to the supplementary material

3. Response to comments on Fig9 can be incorporated into the figure legend

7. PLOS authors have the option to publish the peer review history of their article (what does this mean? ). If published, this will include your full peer review and any attached files.

**Do you want your identity to be public for this peer review?** For information about this choice, including consent withdrawal, please see our Privacy Policy .

Reviewer #1: No

Reviewer #2: No

---

## [Author Response · Author response to Decision Letter 2]

18 Apr 2025

Response to Reviewer Comments

Manuscript Title: Role of Rosuvastatin and Pitavastatin in Alleviating Diabetic Cardiomyopathy in Rats: Targeting of RISK, NF-κB/ NLRP3 Inflammasome and TLR4/ NF-κB Signaling Cascades

Manuscript Number: PONE-D-24-48775R1

Authors: Dalia O. Saleh, Nesma M.E. Abo El Nasr, Marawan A. Elbaset, Marwa E. Shabana, Tuba Esatbeyoglu, Sherif M. Afifi , Ingy M. Hashad

Reviewer #1: I have reviewed the revised manuscript PONE-D-24-48775_R1 and compared it to the initial round of comments. The authors have made significant improvements in addressing the concerns raised in the first review, including clarifying methodologies, refining statistical reporting, improving discussion points, and correcting language inconsistencies. However, some issues remain unresolved or only partially addressed, and these need further attention before the manuscript can be fully accepted.

Respond: We sincerely appreciate the thoughtful and constructive feedback provided by the reviewers. Your insights have greatly enhanced our manuscript, and we are grateful for your time and consideration.

Reviewer's Responses to Questions

Remaining Issues That Require Further Revision

1. Statistical Clarity and Effect Size Reporting (Lines 130–141)

-Unresolved Issue: While the authors have reported statistical significance (P-values), they have still not provided effect sizes (e.g., Cohen’s d, η², or odds ratios for logistic regression models), which are crucial for evaluating biological relevance.

-Suggested Fix: Include effect size measures alongside P-values to quantify the magnitude of observed differences.

Respond: The effect size was added within the supplementary data for each figure and expressed as r2 or eta-squared (η²). Also, we referred to the availability for this table in the statistical section in the manuscript.

As follows:

Parameter Effect size (eta-squared (η²).

Total Cholesterol 0.9174

Triglycerides 0.8620

Glucose 0.8719

RR Interval (s) 0.8886

Heart Rate (BPM) 0.9181

PR Interval (s) 0.4448

QRS Interval (s) 0.3874

QTc (s) 0.9548

R Amplitude (mV) 0.6970

ST Height (mV) 0.4609

NLRP3 (ng/mg protein) 0.9515

Cardiac MDA 0.9349

Cardiac GSH 0.7327

Cardiac pro-fibrotic IL-1 β 0.9410

cardiac Akt 0.8627

p-GSK-3β 0.8346

Cardiac NF-κB 0.9833

TLR-4 0.9390

cardiac troponin 0.9591

2. Overinterpretation of Causal Relationships (Lines 142–200)

- Unresolved Issue: Despite some improvements, the discussion still implies causation in certain statements regarding statins' effects on NF-κB and NLRP3 inflammasome signaling.

-Suggested Fix: Replace phrases like "Statins reduce NLRP3 activation, leading to improved cardiac function" with "Statins are associated with reduced NLRP3 activation, which may contribute to improved cardiac function."

Respond: Done

3. Insufficient Acknowledgment of Confounding Factors (Lines 201–216)

- Unresolved Issue: The authors have not sufficiently discussed how patient heterogeneity, pre-existing inflammation, or medication history might influence their findings.

-Suggested Fix: Add a brief acknowledgment of potential confounders and recommend future studies to control for these variables.

Respond: We appreciate the reviewer’s concern regarding the acknowledgment of confounding factors such as patient heterogeneity, pre-existing inflammation, and medication history. However, we would like to clarify that our study was conducted as a preclinical experiment using animal models (rats) and not human patients. Consequently, the factors of patient heterogeneity and medication history are not directly applicable in our study.

In our preclinical model, we controlled confounding variables through inclusion/exclusion criteria for the animals according to serum glucose levels in the induction of diabetic model. Additionally, pre-existing inflammation was considered by the inclusion of normal rats in the beginning of the experiment and control groups in the experimental design. While these factors may be relevant in clinical settings, they do not directly influence the interpretation of our findings within the context of our preclinical design. We hope this clarification addresses the reviewer’s concerns. Thank you for your thoughtful feedback.

4. Lack of Clarity on Missing Data Handling (Lines 100–129)

- Unresolved Issue: The manuscript still does not specify how missing values were handled in statistical analyses. Were missing values imputed or excluded?

- Suggested Fix: Clearly state the approach used (e.g., multiple imputation, listwise deletion, or mean substitution).

Respond: This sentence was added to the manuscript: “If there are any missing data before analysis, we handled using listwise deletion, wherein cases with missing values were excluded from the respective analyses.”

5. Ambiguous Language and Grammar Issues

- Many language issues have been addressed, but some awkward phrasings remain.

- Example: Line 31: "The Dried Blood Spot (DBS) technique using filter papers is an easy, quick, transport-ease technique..."

- Fix: "The DBS technique is a rapid and convenient micro-sampling method using filter papers."

Respond: We revised the whole manuscript for any ambiguous terms.

Reviewer #2: A. The authors have adequately addressed almost all raised points, only a few were partially resolved:

1. the frequency of the daily drug dose is still not explicitly stated (is the drug administered once or twice daily)

Respond: We appreciate the reviewer’s observation regarding the frequency of drug administration. To clarify, the drug was administered once daily for four consecutive weeks.

We have revised the manuscript to explicitly state this information in the relevant section In abstract: the RVS group of DCM-induced rats that were treated once daily with 10mg/kg of RVS, and the PTS group of DCM rats that were treated with 0.8 mg/kg of PTS).

In materials and methods section:

2.4. Study design:

Rats confirmed to have diabetes were randomly assigned into three groups: an untreated diabetic cardiomyopathy (DCM) group, a DCM group treated orally with (RVS, 10 mg/kg) or (PTS, 0.8 mg/kg) once daily for four weeks. In addition to another group being fed a standard diet with free access to water which served as a normal control. At week seven, these rats received a citrate buffer injection (1 ml/kg, i.p.)

Thank you for pointing this out, and we hope this clears up any confusion.

2. the full definition of the term DMC should be introduced early on in the manuscript

Respond: The abstract and introduction sections now begin with this term (Diabetic cardiomyopathy (DCM))

B. Some of the responses will enhance the clarity of the manuscript if incorporated where appropriate into the main text or added to the supplementary material:

1. The response to "Is a single reading taken one week after STZ administration sufficient...."

Respond: We would like to clarify that blood glucose levels were measured at two time points:

The first measurement was taken one week after STZ administration and served only as an inclusion criterion to confirm the successful induction of diabetes and to select diabetic rats for the study.

The second measurement was taken at the end of the experiment to evaluate the persistent hyperglycemia in the diabetic group and assess the effect of the treatments.

We have added to the manuscript’s main text (Methods section, under Experimental Design/Animal Grouping):

"Blood glucose levels were measured at two time points: initially, one week after STZ administration, to confirm diabetes induction for group allocation; and again at the end of the experimental period to assess the maintenance of hyperglycemia and treatment effects."

2. MT staining steps can be added to the supplementary material

Respond: We have added as Supplementary File 1:

MT staining steps can be added to the supplementary material.

We have prepared a detailed, step-by-step Masson's Trichrome staining protocol to be included in the Supplementary Material for full transparency and reproducibility.

3. Response to comments on Fig9 can be incorporated into the figure legend

Respond: We revised the legend of Figure 9 to include an explanatory note based on the reviewer’s comments, providing clarity on how the measurements were obtained and interpreted.

We added the following part to the legend of Figure 9:

"Figure 9 illustrates the histological examination of myocardial tissue, including evaluation of inflammatory cell infiltration, fibrosis, and fat accumulation characteristic of DCM. The observed changes were semi-quantitatively assessed and supported by Masson’s Trichrome staining for collagen deposition. Treatment with RSV showed partial improvement, while PTS treatment more effectively preserved myocardial architecture, reduced fibrosis, and restored myocyte size closer to the control group values."

---

## [Editor Report · Decision Letter 2]

Role of Rosuvastatin and Pitavastatin in Alleviating Diabetic Cardiomyopathy in Rats: Targeting of RISK, NF-κB/ NLRP3 Inflammasome and TLR4/ NF-κB Signaling Cascades

PONE-D-24-48775R2

Dear Dr. Esatbeyoglu,

We’re pleased to inform you that your manuscript has been judged scientifically suitable for publication and will be formally accepted for publication once it meets all outstanding technical requirements.

Kind regards,

Doa'a G. F. Al-u'datt

Academic Editor

PLOS ONE
---

## [Editor Report · Acceptance letter]

PONE-D-24-48775R2

PLOS ONE

Dear Dr. Esatbeyoglu,

I'm pleased to inform you that your manuscript has been deemed suitable for publication in PLOS ONE. Congratulations! Your manuscript is now being handed over to our production team.

Kind regards,

on behalf of

Dr. Doa'a G. F. Al-u'datt

Academic Editor

PLOS ONE